



# Relationship of seasonal variations in drip water $\delta^{13}C_{DIC}$, $\delta^{18}O$ and trace elements with surface and physical cave conditions of La Vallina Cave, NW Spain

Oliver Kost[1], Saul González-Lemos[2], Laura Rodriguez-Rodriguez[3], Jakub Sliwinski[4], Laura Endres[1], Negar Haghipour[1,5] and Heather Stoll[1]

[1] Geological Institute, ETH Zürich, Sonneggstrasse 5, 8092 Zurich, Switzerland
[2] ASCIEM Consulting SLP; C/ Gutiérrez Herrero 52, 33402 Avilés, Spain
[3] Departamento Geología Universidad de Oviedo, C/ Jesús Arias de Velasco s/n, 33005 Oviedo, Spain
[4] School of Earth and Environmental Sciences, University of St Andrews, Queen's Terrace, KY16 9AJ, St Andrews, Scotland
[5] Ion beam Physics, ETH Zürich, Otto-Stern-Weg 5, 8093 Zurich, Switzerland

Correspondence to: Heather Stoll (heather.stoll@erdw.ethz.ch)

**Abstract.** Cave monitoring studies clarify the climatic, surface vegetation, and karst processes affecting the cave system and lay the foundation to interpreting geochemical stalagmite records. Here we report monitoring of cave air, bedrock chemistry, and drip water $\delta^{13}C_{DIC}$, $\delta^{18}O$ and $\delta D$ as well as 16 trace elements covering a full annual cycle spanning 16 months between November 2019 and March 2021 in La Vallina cave in the Northwest Iberian Peninsula. While decreased rainfall and increased evapotranspiration in summer months lead to a strong reduction in drip rates, there is little seasonal variation of $\delta^{18}O$ and $\delta D$ in a given drip, likely reflecting discrete moderately- to well-mixed karst water reservoirs. Small differences in $\delta^{18}O$ and $\delta D$ between drip sites are attributed to variable evaporation intensity and/or transit times. The dissolved inorganic carbon composition of drip water ($\delta^{13}C_{DIC}$) is likely driven by seasonal changes in temperature controlling biological processes (vegetation and microbial soil activity) resulting in minimum $\delta^{13}C_{DIC}$ in summer and autumn months. Increased bedrock dissolution due to higher soil $pCO_2$ in summer and autumn results in increased trace element concentrations of congruently dissolved elements. Cave air measurements indicate seasonal ventilation (winter) and stagnation (summer) of cave air. The opposite effects of reduced cave air $pCO_2$, seasonally variable biological activity and increased drip rate limit the extent of seasonal variation of degassing and prior calcite precipitation (PCP) supported by trace elements (Sr/Ca-index). Estimated stalagmite growth rates using monitoring data suggest calcite precipitation is restricted to certain seasons (summer and winter) at certain locations within the cave, which has important implications on proxy interpretation of stalagmite records.

## 1. Introduction

While speleothems represent long-lasting paleoclimate archives with a host of potential elemental and isotopic proxies for climate, the interpretation of these proxies depends on the understanding of complex physical, chemical and biological



processes. Because of ongoing improvements in the sensitivity of inductively coupled plasma mass spectrometry (ICP-MS), and *in situ* techniques such as laser ablation (LA)-ICP-MS, a wider range of trace elements can be analyzed at unprecedented precision and spatial resolution, shedding light on seasonal and sub-annual variations in trace element abundance in stalagmites (e.g. Faraji et al., 2021; Sliwinski and Stoll, 2021). For this reason, the limiting factor in interpreting trace element signals in

speleothems is no longer analytical instrumentation, but rather our understanding of underlying physical processes. Cave monitoring studies provide one important approach for improving the understanding of speleothem proxies, because measurements of elemental or isotopic composition of drip water can be compared with instrumental measurements of seasonal or interannual variations in parameters of climatic interest, such as infiltration rate, rainfall amount and origin, and temperature (e.g. Breitenbach et al., 2015; Hartland et al., 2012; Riechelmann et al., 2011; Spötl et al., 2005). Cave drip waters represent a

unique window on the flow rates and biogeochemistry of infiltrating groundwater in karst regions, providing information on parts of the system which are otherwise difficult to sample.

Speleothem studies have traditionally relied heavily on the interpretation of $\delta^{18}O$, and global monitoring comparisons have confirmed that drip water $\delta^{18}O$ generally follows rainwater compositions (Baker et al., 2019). However, seasonal changes in rainwater $\delta^{18}O$ can appear with significant lag and attenuation by the karst reservoir (e.g. Moreno et al., 2014; Markowska et

al., 2016). Evaporative fractionation of karst water in the vadose zone affects the water isotopic composition and can even exceed the rainwater imprint (Baker et al., 2019; Markowska et al., 2016). The potential for differences in the importance of recharge from different seasons leads to $\delta^{18}O$ differences among coeval stalagmites (e.g. Bradley et al., 2010) in contrast to evaporative and degassing effects by Prior Calcite Precipitation (PCP; Deininger et al., 2021) and remains to be resolved through further monitoring studies.

Varying trace element ratios in speleothems have most commonly been attributed to the effects of PCP for divalent cations like Mg, Sr, and Ba (e.g. Fairchild and Treble, 2009; Sinclair et al., 2012), and on the effect of organic chelation on the colloidal transport of some insoluble metals (e.g. Cu, Ni, Zn; Hartland et al., 2012; Borsato et al., 2007). Drip water monitoring has demonstrated PCP processes in several cave systems (e.g. Fairchild et al., 2000; Riechelmann et al., 2011; Sinclair et al., 2012) and shown that seasonal cave ventilation, as well as drip rate, can regulate the degree of PCP occurring in drip waters (Mattey

et al., 2010). However, while many monitored drip waters often show covariation in Mg/Ca, Sr/Ca, and Ba/Ca expected from PCP, these ratios often do not covary in coeval stalagmites in ways expected from PCP (Sinclair et al., 2012; Stoll et al., 2012). Such deviations may be due to variations in partitioning coefficients of some trace elements (e.g. Stoll et al., 2012; Wassenburg et al., 2020) or may reflect heterogeneity in Mg content and solubility of different phases in the host rock (e.g. Tremaine et al., 2016). Drip water monitoring can be used to ascertain the significance of this latter process.

Transition metals, P and Y are increasingly reported in speleothems, and typically related to colloidal transport processes, but the very low concentrations of these elements in drip waters has meant that very few studies have reported monitoring data (Baldini et al., 2012; Hartland et al., 2012). Seasonal peaks in Y in stalagmites have been attributed to periods of high discharge at the end of the growing season in autumn (Borsato et al., 2007), yet a high-resolution drip water monitoring study found highest Y concentrations in dry late summer months, potentially due to significant dry dust deposition in the collection vials



residing in the cave (Baldini et al., 2012). Therefore, it remains uncertain if speleothem content of these insoluble elements is controlled exclusively by drip water deposition, and if flux in water is dependent only on hydrological factors or also on the factors regulating biological production of chelating ligands.

Carbon isotopic ratios are measured in stalagmite samples synchronously with $\delta^{18}O$ but have been less frequently interpreted as a proxy because they are affected by both soil/vegetation processes and degassing of drip waters prior to and during

speleothem deposition. Their effect on $\delta^{13}C$ can thus be difficult to deconvolve (e.g. Fohlmeister et al., 2020; Frisia et al., 2011; Lyu et al., 2020). Monitoring the $\delta^{13}C$ of dissolved inorganic carbon ($\delta^{13}C_{DIC}$) in drip water would be useful to evaluate these processes but has been limited (e.g. Fohlmeister et al., 2020; Frisia et al., 2011; Spötl et al., 2005) and may be affected by variable and extensive degassing during sample collection. Some studies report analyses of $\delta^{13}C_{DIC}$ on samples which have resided for weeks to months in the cave and therefore been subject to very long exchange and degassing times and may not be

representative of the $\delta^{13}C_{DIC}$ involved in speleothem deposition. Most carbon isotope interpretation is therefore based on theoretical models (Deininger and Scholz, 2019) with limited observational comparison.

Here we evaluate the response of drip water stable isotopes and trace elements to variations in ventilation, temperature, hydrological variability and local vegetation activity through a 16 month monitoring study in La Vallina Cave, NW Spain. This cave is the site of a number of paleoclimatic records reconstructed from stalagmites (e.g. Sliwinski and Stoll, 2021; Stoll

et al., 2015; Stoll et al., 2013), and features seasonal climate cycles and cave ventilation dynamics which are expected to be representative of many midlatitude systems. We demonstrate that $\delta^{13}C_{DIC}$ of drip water varies seasonally and is tightly linked to variations in vegetation and soil respiration processes primarily controlled by ambient temperature. Cave air $CO_2$ data help to understand cave ventilation processes and can constrain the $\delta^{13}C$ of the respired soil endmember. Further, we document seasonal differences in the intensity of bedrock dissolution driven by seasonally changing water acidity controlled by soil

$pCO_2$. In this study we analyze 16 elements which are brought into perspective with bedrock dissolution but also other sources are discussed. Furthermore, trace element data help to constrain when prior calcite precipitation (PCP) occurs. Oxygen isotope data suggest multiple discrete moderately- to well-mixed drip water reservoirs fed by rain water, with site-specific differences controlled by evaporation effects. This can affect the interpretation of stalagmite $\delta^{18}O$ signals, suggesting that $\delta^{18}O$ differences in coeval stalagmites do not have to be attributed to kinetic effects during calcite formation, but rather isotopic variability in

the drip water source. Such cave monitoring data sets can help to constrain process models like CaveCalc (Owen et al., 2018) or I-STAL (Stoll et al., 2012). For example, based on cave air and drip water chemistry data, phases of theoretical stalagmite growth can be modeled.

## 2. Cave and sampling sites

La Vallina Cave (43°24'36''N, 4°48'24''W; 70 m asl) is situated in the Carboniferous limestone (Barcaliente Formation;

Gibbons and Moreno, 2002; Álvarez et al., 2019) and belongs to the North Iberia Speleothem Archive (NISA) (Stoll et al., 2015; Stoll et al., 2013) (Fig. 1A). The entrance is situated 70 m above sea level on a northward facing hill slope at a distance





of 2.5 km from the coast, and typical rock thickness above the cave is about 5-30 m (Fig. 1C; Table 1). The vegetation covering the cave varies from oak trees to recently grown eucalyptus and shrub/grass land (Fig. 1B, C). There are also patches used for stock farming (cows, horses and sheep), while sporadic dolines indicate karstification of the hostrock. La Vallina cave consists
of two major galleries on top of one another (Fig. 1B & C).

The nearby weather station of Llanes (43°25'13''N, 4°44'53''W; 10 m asl) 4.75 km northeast from the cave provides weather and climate data with daily and monthly average precipitation and temperature varying between 40-140 mm month$^{-1}$ and 10-20°C respectively (based on years 2000-2020; Fig. 1D; Aemet, 2021). According to the Köppen-Geiger classification, La Vallina cave is situated in a Cfb type climate (Peel et al., 2007). Thus, the NW Iberian Peninsula is characterized as a temperate
oceanic climate with mild winters. Regarding the annual precipitation, there is a water deficit in summer when potential evapotranspiration overcomes precipitation (Fig. 1D). Nevertheless, the aridity index based on the ratio of precipitation to potential evapotranspiration (P/PET) is equal to 2.1, classifying NISA as "humid" according to the UNEP classification (Unep, 1997). The monitored period 2019-2021 was 16% wetter compared to the 20 year average precipitation.

Drip site Gloria is situated 12.8 m beneath shrubby and ferny vegetation cover (Fig. 1 B, C). The drip water samples were
taken from a soda straw on the cave ceiling. Gravel is a zone highly decorated with helictites and eccentric soda straws at the cave ceiling 6.8 m beneath shrubby vegetation cover and nearby trees (oak and eucalyptus) whose roots reach the cave. The location takes its name from evidence of a gravely filled vertical shaft in the overlying bedrock. The sampled soda straw for Gravel was changed over the course of the monitoring period with less than 0.3 m since the originally chosen soda straw dried out and another one broke off. Playground and Playground-01 (starting end of October 2020) are situated a few meters apart
and covered by 24.1 m of bedrock situated beneath grassland used as animal pasture for part of the year. The exceptional drip site Skyscraper is a stalactite with very high flow rate (shower like) suggesting joint flow from a wider area focusing at this stalactite hanging a few meters above ground. Therefore, DIC sampling was not possible since the ceiling was not reachable. Skyscraper is situated at the transition between grassland and bushes with a single oak tree and overlain by 30.7 m of bedrock. Entrada (starting September 2020) and Snowball (from January 2021) were added to the sampling sites later in the study to
capture drip water from a more tree-covered area and thinner (ca. 7.0 m) bedrock cover. These latter locations are nearer to the cave entrance and rapidly exchange cave air with atmosphere.



**Fig. 1: The monitoring site situation. A) Northern Iberian Speleothem Archive (NISA), B) La Vallina Cave with two cave galleries shown on an aerial image and a DEM from 2017 (Instituto Geográfico Nacional, 2017) C) Cross section (X-X') of La Vallina Cave with projected sampling sites, D) climogram from Llanes weather station with 20 year (2000-2020; data from Aemet, 2021) average temperature (red), precipitation (blue bars) and monthly potential evapotranspiration (PET, green) for 2020. Maps were generated with ArcGIS using publicly available data from the Spanish Instituto Geográfico Nacional (2017).**


## 3. Methods

Over the course of 16 months (November 2019 until March 2021) a complete seasonal cycle was monitored at the four
locations (named Gloria, Gravel, Playground and Skyscraper), except that no samples could be taken between February 24th
2020 and May 5th 2020 due to lockdown restrictions of the COVID-19 pandemic. In the second half of the monitoring period,
additional locations were added (Entrada, Playground-01, Snowball; see sample description). In autumn 2020, a biweekly
routine was intended to catch the onset of the rainy season accompanied with colder temperatures affecting cave air stability.
All samples were collected instantaneously in a single 1-30 minute period, so that variations in conditions between sampling
days is not reflected in the drip water chemistry. This strategy, rather than aggregate bottle collection of water samples, was
adopted to obtain optimal geochemical data on parameters which evolve during long collection periods in the cave such as
$\delta^{13}C_{DIC}$ (degassing) and Ca concentration (calcite precipitation) or dust deposition. All water samples reflect collection from
the same soda straw drip source, except at the location Gravel, where the same soda straw was not always active and was
substituted for an adjacent one on different samplings. Cave temperature and relative humidity were recorded at each sampling
site (Table 1). Cave air samples were collected for $pCO_2$ and $\delta^{13}C_{air}$, aliquots of drip water were sampled for geochemical
analysis ($\delta^{13}C_{DIC}$, $\delta D$ and $\delta^{18}O$ and trace elements), and drip rates were recorded by counting drips to constrain hydrological
variation. Meteorological data (temperature, precipitation) were obtained from the nearby meteorological station in Llanes
(Aemet, 2021).

Cave air was collected using a 200 mL syringe (human breath influence avoided) and injected into 0.6 L Supel™-Inert Multi-
Layer Foil aluminum gas sampling bags. The carbon isotopic composition and $pCO_2$ of cave air was measured on a cavity
ringdown spectrometer (CRDS) Picarro G2131-i Isotopic $CO_2$ instrument at ETH Zürich. To apply an offline calibration two
concentration reference gasses (399.6 ppmV and 2,000 ppmV $CO_2$ in synthetic air) and two additional isotope reference gasses
($\delta^{13}C$ = -27.8 and -2.8‰ VPDB) were used.

For drip water $\delta^{13}C_{DIC}$ measurements, drip water was collected directly from soda straw stalactites on the cave ceiling using a
1 mL syringe to minimize degassing effects. Additionally, to evaluate the degassing and equilibration evolution of drip water
exposed to cave air a few samples along the flow path on a flowstone were collected. The sampled water was injected into He
flushed glass vials (Labco Exetainers®) prepared with 150 µL pure phosphoric acid following the procedure of Spötl et al.
(2005). Samples were shipped to ETH Zürich and measured within a week of collection on a Thermo Fisher Scientific Gas
Bench II, equipped with a CTC autosampler and coupled to a ConFlow IV interface and a Delta V Plus mass spectrometer.
The $\delta^{13}C_{DIC}$ measurement uncertainty (1 SD) was typically in a range of ±0.18‰ or lower. Two in-house calibrated sodium
bicarbonate powders (-4.66 and -7.94‰ VPDB) were used for standardization. Linearity and signal intensity effects were
considered.

A handful of drip waters were analyzed for radiocarbon with the Micadas gas ion source at ETH Zürich (Ruff et al., 2007).
Samples were purged with He (99.96% purity) under a flow of 80 mbar min[-1]. Then, 200 µl of $H_3PO_4$ (85%) was added to the
Labco vials and the $CO_2$ was introduced to the gas ion source using a carbonate handling system (CHS). The IAEA C1 ([14]C





dead carbonate standard) was used as blank material and NIST SRM 4990C oxalic acid (OXAII) was used for standard normalization and to check the stability of the measurement. A coral standard ($F^{14}C = 0.9447$) was used as reference material. If we assume that the freshly respired soil carbon has $F^{14}C = 1$ (fully modern), then the dead carbon fraction (DCF) can be estimated by:

$$DCF = 1 - F^{14}C$$

Since after percolating through the epikarst and dissolution of bedrock containing dead carbon the $F^{14}C$ of drip water is reduced resulting in a higher DCF. We evaluate this assumption further in the discussion.

The oxygen and hydrogen isotope composition of drip water was determined from drip water aliquots directly sampled in 3.5 mL Labco vials. By recording the time until the vial was filled, a drip rate (mL min$^{-1}$) was determined. The samples were

measured on a Picarro L2130-I Isotopic $H_2O$ vaporization instrument at ETH Zürich. A sample-standard bracketing method was applied using three in-house water standards (Mediterranean Sea water: 11.8‰ δD, 2.01‰ δ$^{18}$O; Lake Zürich water: -76.4‰ δD, -10.94‰ δ$^{18}$O; Siberian water: -264.7‰ δD, -33.95‰ δ$^{18}$O) which were calibrated relative to IAEA certified standards (SLAP2, GISP and VSMOW2). Seven replicates were measured for each sample, where only the last four were used to determine the isotopic composition to minimize memory effects. Measurement uncertainty (1 SD) is lower than 0.10‰ for

δ$^{18}$O and 1.15‰ for δD. We compare drip water variation with published rain isotope analyses for previous years (Moreno et al., 2021), since no GNIP monitoring at nearby stations is available for the interval of our drip water monitoring.

For elemental analysis, drip water was collected directly in 2 mL Eppendorf tubes which were previously cleaned in a 5% $HNO_3$ acid bath and rinsed several times with Milli-Q®-water to minimize vial contamination. To avoid contamination from metal syringes used in DIC sampling, water for these samples was allowed to drip from the soda straw directly into the tube.

Unlike the DIC sample, the water therefore experienced some PCP at the tip of the soda straw. To avoid in-vial calcite precipitation, tubes were filled in the cave with no headspace and sealed. In the lab, drip water was centrifuged to separate any particulate matter, after which 500 µL of supernatant were transferred into Teflon vials and acidified with 15 µL high purity (double distilled) 65% $HNO_3$ to bring the sample to match the matrix of the standards (2% $HNO_3$). This reduces clogging of the ICP-MS nebulizer related to water surface tension and prevents any in-vial calcite precipitation.

In-house trace element standards were prepared using certified mono-element standards (ISO 17034 CRM by INORGANIC VENTURES) mixed in a 2% $HNO_3$ matrix to yield multi-element calibration standards covering the range of drip water chemistry. Drip water trace elements were measured on an Agilent 8800 QQQ-ICP-MS at ETH Zürich. The following list of 16 elements are well above the level of quantification (LOQ): Li, Na, Mg, Al, Si, S, K, Ca, Cr, Mn, Cu, As, Sr, Y, Ba and U (Table A1). Although K is seldom reported in trace element studies due to tailing of the ICP carrier gas ($^{40}Ar$) on $^{39}K$ and

interference with $^{38}Ar^1H$, analysis in collision mode with $H_2$ reduces the Ar tailing so that measured $^{39}K$ intensities are stable and several orders of magnitude above LOQ. Likewise, intensities of $^{25}Mg$ and $^{55}Mn$ are measured in $H^2$ collision mode to improve measurement stability and to avoid $^{40}Ar^{15}N^+$ interferences, respectively. Helium collision mode gives most stable measurements for $^{23}Na$ by reducing the $^{46}Ca^{++}$ interference. $^{32}S$ and $^{29}Si$ are mass shifted in $O_2$ reaction mode (+16 mass units) to reduce interferences by $O_2$ and $N_2$, respectively. Other elements are measured without any collision/reaction gasses.



Measurement uncertainties are typically in the range of a few percent relative standard deviation (RSD; three replicates). The range of elemental concentrations spans from low pptV (pg g$^{-1}$) levels (e.g. Y, Cr or Cu) up to tens of ppmV (µg g$^{-1}$) levels (Na, S, Mg, Si or Ca).

Rock samples were collected from fallen blocks of limestone on the floor throughout the cave and in some locations from the cave wall (Table A2). From each hand specimen, powder was extracted with a 1 mm dental drill and homogenized before

dissolution in 2% HNO$_3$ to analyze trace elements with an Agilent 8800 QQQ-ICP-MS at ETH Zürich using intensity ratio calibration (De Villiers et al., 2002) and in-house multi-element standards of varying trace element-to-Ca ratios, but with Ca concentrations matched to samples (400 ppmV). Aliquots of the rock powder were measured for stable isotopes ($\delta^{13}$C and $\delta^{18}$O) on a Thermo Fisher Scientific Gas Bench II, equipped with a CTC autosampler and coupled to a ConFlow IV interface and a Delta V Plus mass spectrometer. Isolab B and MS2 (carbonate reference material) were used for standardization yielding

uncertainties typically less than 0.1‰.

Principal component analysis (PCA) was applied using MATLAB after applying a z-score normalization to elemental concentrations of individual drip sites. Elements yielding sub-LOQ values were defined as zero to compute the PCA. This multi-dimensional correlation analysis approach identifies common modes (principal components: PC 1, PC 2, …, PC $n$) controlling the chemical signal and allows the grouping of elements with similar behaviour. Similar analysis has previously

been used in speleothem studies (e.g. Borsato et al., 2007; Fairchild et al., 2010; Treble et al., 2016).

We estimate relative variations in PCP in each drip site prior to our drip water collection using the Sr/Ca ratio, since we infer that both elements (Sr and Ca) are primarily bedrock dissolution controlled. To avoid potential influence of marine salt contributions in this ocean proximal cave we intentionally do not use the more commonly used Mg/Ca ratio as a PCP proxy. Assuming congruent dissolution, the Sr/Ca ratio increases with increasing Ca loss due to PCP. As a relative index of PCP, we

calculate the following Sr/Ca-index:

$$\text{Sr/Ca-index} = (\text{Sr/Ca})_{min} / (\text{Sr/Ca})$$

(Sr/Ca)$_{min}$ of each specific drip site is the least affected by PCP and is defined as having a Sr/Ca-index of one. Thus, the Sr/Ca-index describes the inverse of the PCP effect or namely the percentage of residual Ca. For example, a value of 0.8 means that

80% of the reference Ca remains in solution and 20% is lost to PCP.

We employ the CaveCalc model (Owen et al., 2018) to explore the processes responsible for $\delta^{13}$C$_{DIC}$ control. By comparing simulations and monitored data we can rule out potential drivers controlling $\delta^{13}$C$_{DIC}$. Furthermore, we applied the I-STAL model (Stoll et al., 2012) using drip water Ca concentrations, cave air conditions (pCO$_2$ and temperature) and drip intervals to simulate a full annual cycle of stalagmite growth at the perennially monitored sites. I-STAL simulates stalagmite growth using

the kinetic model of Dreybrodt (1999), which is based on the oversaturation defined by the difference between measured Ca concentration and the Ca concentration at equilibrium with CaCO$_3$ at cave temperature and atmosphere. The resulting average deposition rate of calcite was converted into an instantaneous vertical growth rate (µm yr$^{-1}$).





# 4. Results

## 4.1 Cave air

During the monitored period, the relative humidity is always higher than 90% (Table 1). Close to the cave entrance (Entrada) exchange with ambient air can affect both humidity and temperature. The cave temperature shows very limited seasonal variation (Fig. 2a). At monitored locations deep in the cave the temperature varies less than 1°C at a mean temperature of 13.9°C (Table 1), which is slightly colder than the 20-year average temperature of Llanes weather station (14.7°C). Closer to the cave entrance, where exchange with outside air is greater (Entrada, Snowball and Gloria), temperature ranges from 13.5 to

15.2°C. Temperature was only recorded from June 2020 to March 2021, but we assume similar stable cave temperatures from the start of the monitoring period in all locations except Entrada (dashed line in Fig. 2a).

Cave air $CO_2$ concentration ($pCO_2$) varies seasonally, increasing in spring, and remaining high in summer and autumn months before dropping to ambient air levels in winter. The first $pCO_2$ decrease in both years coincides with outside air temperatures dropping below cave air temperature, suggesting a relationship with cave ventilation (Fig. 2a and d). Stratified cave air

conditions are expected during summer months when outside temperature is warmer than cave air. When the atmospheric air temperature is similar or lower than cave air temperature, the cave air is exchanged with outside air due to temperature controlled density differences. The cave air exchange in autumn differs among the two years in some sites. At Gravel and Skyscraper, $CO_2$ increased and reached its highest $pCO_2$ in November and December 2020 following an initial drop in September and October 2020. Coincident with differing external air temperature evolution, the cold season $pCO_2$ decline was

earlier in 2019 than 2020. In 2019, for example, $pCO_2$ reached maximal values of 1,320 ppmV in November compared to >5,000 ppmV in November 2020. Between December 2019 to February 2020 and December 2020 to January 2021, the cave air $pCO_2$ dropped to almost outside air conditions (black line in Fig. 2d, sampled in the surrounding forest). To bring cave $pCO_2$ into context with outside air temperature, more meteorological data before the start of the monitoring period would be beneficial. Unfortunately, the weather station at Llanes did not generate any data between July and October 2019. Different

maximal $pCO_2$ levels are observed at various elevations within the cave. Topographically lower positions in the cave accumulate $CO_2$ and concentrations reach 4,000-5,400 ppmV (Skyscraper and Playground). Topographically higher portions of the cave feature maximal $pCO_2$ of 1,500-2,500 ppmV (Gravel, Gloria and Entrada).

A seasonal structure is also evident in the carbon isotopic composition of cave $CO_2$ ($\delta^{13}C_{air}$; to distinguish from $\delta^{13}C_{DIC}$). While the measured air *outside* the cave remains fairly constant over the year (average of -11.6‰, black line), cave air evolves to

very negative $\delta^{13}C$ values in spring and remains very negative during summer months. The most negative $\delta^{13}C_{air}$ values are reached at locations with high $pCO_2$ deeper in the cave (Skyscraper: -26.0‰ in May 2020). The $\delta^{13}C_{air}$ approaches outside air conditions between December and February (2020) or January (2021) in both monitored winters, in line with the cave air $CO_2$ concentration decrease. The autumn transition differs among the sites with only Gloria and Entrada indicating an intermediate shift closer to atmospheric values at the end of October (2020) and other sites diverging from atmospheric $\delta^{13}C_{air}$ composition

to reach another extreme in negative composition (e.g. Gravel, Skyscraper).



## 4.2 Dissolved inorganic carbon (DIC) of drip water

Drip water from within soda straws growing on the cave ceiling were sampled regularly to measure the changes in carbon isotopic composition of drip water DIC ($\delta^{13}C_{DIC}$) over time. Substantial seasonal variations in the $\delta^{13}C_{DIC}$ measurements are observed, with the most negative values (-17.0‰) observed in summer and autumn months. In winter, the $\delta^{13}C_{DIC}$ generally

increases (to -8.4‰ at Playground) (Table 1, Fig. 2b), and the maximum difference between the two extreme values can reach 8.6‰ (Playground; red in Fig. 2). Large seasonal variations (7.7‰) are also observed for Gravel (green) with the most negative values in late autumn (mid November). Smaller variations between summer and winter are recorded at Gloria (blue) where $\delta^{13}C_{DIC}$ varies within a range of 4.0‰ (-15.5 to -11.5‰).

**Table 1: Cave environment measurements at monitoring locations giving ranges (min-max) or mean values and variations (%-RSD).**
**The full data set of temperature, cave air parameters (pCO$_2$ and $\delta^{13}C_{air}$) and drip water DIC measurements ($\delta^{13}C_{DIC}$) are plotted in Fig. 2. The full time series of drip rate and $\delta^{18}O$ are plotted in Fig. 3.**

| Location | Bedrock thickness | Rel. humidity [%] | Temperature [°C] | | pCO$_2$ [ppmV] | | $\delta^{13}C_{air}$ [‰ VPDB] | | Drip rate [mL min$^{-1}$] | | $\delta^{13}C_{DIC}$ [‰ VPDB] | | $\delta^{18}O$ [‰ VSMOW] | |
|---|---|---|---|---|---|---|---|---|---|---|---|---|---|---|
| | [m] | Min | Min | Max | Min | Max | Min | Max | Mean | %-RSD | Min | Max | Min | Max |
| Gloria | 12.8 | >92.4 | 14.1 | 14.9 | 569 | 1582 | -21.2 | -14.1 | 0.27 | 54 | -15.5 | -11.5 | -6.7 | -5.6 |
| Gravel | 6.8 | >92.7 | 13.5 | 14.0 | 509 | 2416 | -23.0 | -14.1 | 0.67 | 50 | -16.2 | -8.5 | -6.5 | -5.1 |
| Playground (Playground-01*) | 24.1 | >90.9 | 13.5 | 14.0 | 525 | 4199 | -24.4 | -15.4 | 1.48 (0.33) | 54 (21) | | | -6.5 (-6.3) | -5.8 (-5.4) |
| Skyscraper | 30.7 | >94.5 | 13.6 | 14.2 | 510 | 5390 | -25.6 | -14.0 | 91.72 | 30 | - | - | -6.7 | -5.8 |
| Entrada* (Snowball*) | 7.0 | >91.5 | 13.5 | 15.2 | 461 | 1496 | -21.0 | -12.1 | 0.38 (0.36) | 46 (83) | -16.6 (-13.8) | -14.0 (-12.9) | -7.1 (-6.8) | -5.2 (-5.6) |

*Not complete seasonal cycle

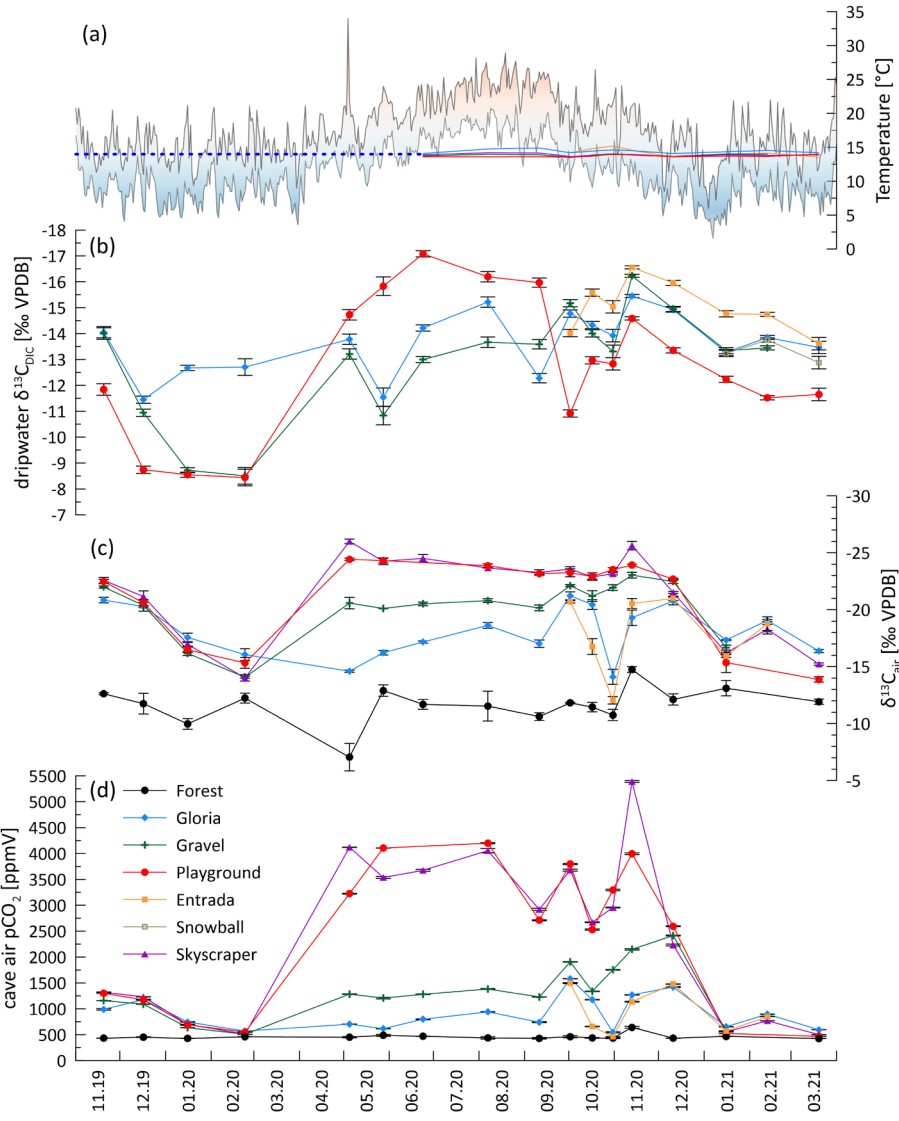

**Fig. 2: Cave air conditions and drip water DIC over time. a) Daily outside temperature range (min-max) in Llanes (Aemet, 2021) and cave air temperature. The blue dashed line extrapolates the constant cave air temperature. b) drip water $\delta^{13}C_{DIC}$, c) cave air $\delta^{13}C_{air}$ and d) cave air $pCO_2$. Note the reversed y-axes for B and C.**

## 4.3 Hydrological conditions

The drip rate of all the monitored sites varies seasonally, though there are large differences (several orders of magnitude) in the absolute drip rates across the various drip sites (Table 1, Fig. 3d). Therefore, a normalization was applied to directly compare the different flow conditions compared to the maximal observed individual flow (Fig. 3d, e). In summer, most drip sites drop to 10-25% of its highest discharge recorded in the wet season, consistent with the lower rainfall (P) and higher temperatures which maximize evapotranspiration (PET; Fig. 1d). Recharge of the water reservoir potentially only occurs when



(P-PET) is positive (Fig. 3e). Recently published hydrological models from semi-arid caves suggest a recharge threshold has to be overcome (saturated soil and epikarst) to contribute to the deeper water reservoir feeding the cave drip water (Markowska
et al., 2016; Baker et al., 2021). Hence, when (P-PET) is negative in summer, any rain is likely evaporated and does not recharge the water reservoir. However, persistent drip rates suggest that the water reservoir never dries out (Fig. 3d). However, this interpretation is complicated by seasonal variations in (P-PET), the use of point sampling (as opposed to continuous sampling) and potential variations in the lag between peak precipitation and peak drip rate. Hence, no statistically significant relationship between drip rate and rainfall (e.g. antecedent 7-day cumulative rainfall amount) is found as suggested by other
studies (Baker et al., 2020; Baker et al., 2021). Nevertheless, comparison of Fig. 3e and 3f suggests different timescales of response to recharge (P-PET). For example, Playground and Skyscraper (red and purple respectively) respond rapidly, while Gravel (green) suggests a time lagged drip rate response. However, no statistically significant relationship is found. Although outside the scope of this study, more sophisticated hydrological investigations (e.g. continuous monitoring, tracer study, recharge model) would be beneficial to improve the understanding of the hydrological system.

**4.4 Isotopic composition of the drip water**

Both the oxygen ($\delta^{18}$O) and deuterium ($\delta$D) isotopic compositions of drip water were measured (Fig. 3a and b). The time series of $\delta^{18}$O shows a fairly stable composition for individual drip sites (Fig. 3b). Skyscraper and Playground do not show substantial seasonal trends and record minor $\delta^{18}$O variability within a range of 0.9‰ (-5.8 to -6.7‰) and 0.7‰ (-5.8 to -6.5‰) respectively. Larger variations appear in other drip sites: Gloria varies by 1.1‰ in a range between -5.6 to -6.7‰ with more
negative values at the beginning of the monitoring period. On the other hand, Gravel reveals an opposing trend with variations between -5.1 to -6.5‰, where the most negative values are recorded in winter (20/21). Between Gravel and Gloria, a maximum $\delta^{18}$O offset of 1.5‰ is observed at the beginning of the monitoring phase and converges towards the end. Thus, drip site specific differences in the water pool $\delta^{18}$O are inferred. A similar trend is observed in $\delta$D measurements: Gloria features more positive values in winter 20/21 (-29‰) while Gravel reaches the most negative values during the same winter season (-32‰).
The short time series of Entrada reveals strong variability in both isotopic ratios (-5.2 to -7.1‰ $\delta^{18}$O; -26 to -38‰ $\delta$D). Similarly, the short data sets of Playground-01 and Snowball do not allow the discussion of seasonal variability, but they can nevertheless be used to compare the isotopic composition relative to the longer monitored drip sites. Playground-01 behaves differently compared to the nearby Playground drip site (opposite trend in $\delta^{18}$O and $\delta$D; Fig. 3a, b). Hence, Playground-01 is rather comparable to Entrada and reveals high variability suggesting
more focused hydrological flow conditions. Markowska et al. (2016) have shown that nearby drip sites can undergo very different routing in the unsaturated zone undergoing different evaporative fractionation. In general, relatively constant isotope values are observed through the dry season at individual sites suggesting no change of the individual water reservoir by recharge. All sampling sites (except Playground) follow a trend towards more negative values in the last two months of the monitoring period.

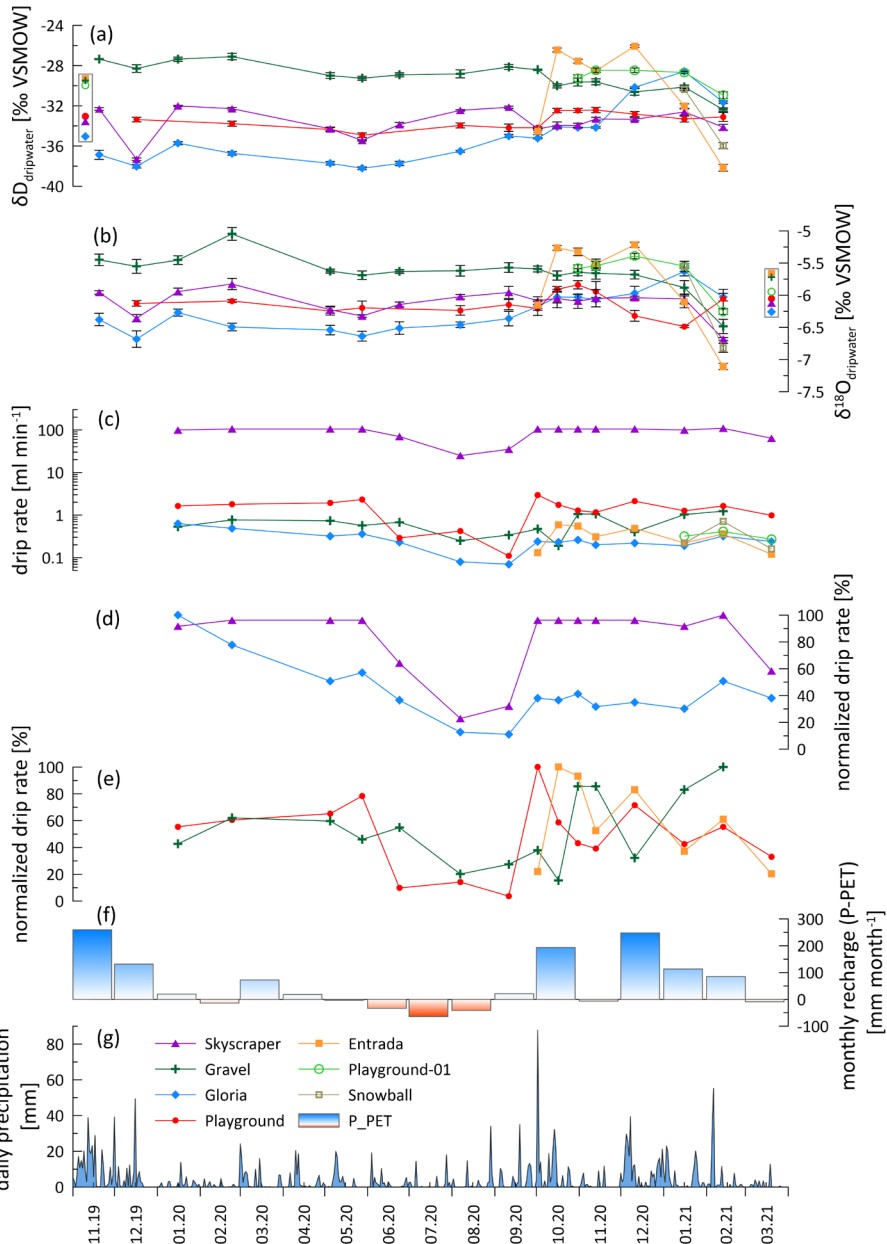


**Fig. 3: Hydrological conditions and water chemistry. a) deuterium isotopic drip water composition (δD$_{dripwater}$), b) oxygen isotopic drip water composition (δ$^{18}$O$_{dripwater}$), c) absolute drip rate (log-scale) d) normalized drip rate of less responsive drip sites (Skyscraper and Gloria), e) normalized drip rate of strongly responsive drip sites with different lags to recharge (Playground, Gravel, Entrada), f) monthly recharge calculated by subtracting potential evapotranspiration (PET) from precipitation (P) and g) daily precipitation**
**recorded in Llanes (Aemet, 2021). In a) and b) Data points in a box next to the y-axis represent amount weighted average values of each drip site.**





### 4.5 Trace elements in drip water and bedrock and temporal variation in PCP

The set of 16 trace elements (TE) analyzed and reported in this study gives a broad view of the seasonal and site specific variability of elemental concentrations in drip water. Fig. 4 only shows TE data above LOQ (Table A1). The elements P, Fe,

Co, Ni, Zn, Cd and Pb did not yield sufficient detectability (mostly below LOQ), hence cannot be treated further in this study. Drip site specific concentration offsets are observed for some elements (e.g. Mg, S, Na, U etc.). Many elements show seasonal variability (e.g. Ca, Si, Sr etc.) whereas others feature event scale peaks (e.g. Mn, Y, Cu etc.) independent of the seasonal cycle.

Principal components analysis (PCA) allows for a more structured analysis of the covariations among the large number of

trace elements analyzed in this study. A PCA was performed for each of the four main drip water monitoring sites (Gravel, Gloria, Skyscraper and Playground), wherein the first three principal components (PC 1-3) explain at least 60% of the signal variation (up to 75% for Gravel) and comprise similar elemental loadings (Fig. 5). PC 1-3 coefficients are controlled by similar groups of elements with a consistency that suggests they are controlled by the same environmental factors (Fig. 5). Elements Na, Mg, Si, Ca, Sr, Ba and U are strongly controlled by PC 1 (28-40% of total variation). PC 2 explains about a fifth (18-22%)

of the trace element signal variation and is controlled by elements such as Li, Al, K, Cr, Mn, Cu, As and Y. The third component (PC 3) accounts for 10-18% of the elemental concentrations, and is most associated with S, Cr, Cu and As. While there are exceptions to these rules (e.g. poor associations of PC 2 with Li at Skyscraper, PC 2 with Cu at Gravel and PC 3 with Cu at Gloria and Skyscraper), and discussing every single element at individual drip sites would go beyond the scope of this study, we nevertheless attempt to provide a general interpretation where at least two drip sites show strong element specific PC's.

Increased PCP, indicated by a lower Sr/Ca-index, occurs in autumn and early winter (Fig. 6b). Gravel and Gloria exhibit the strongest variation in the degree of PCP with Sr/Ca-index values decreasing down to 0.60 indicating 40% loss of Ca relative to the reference. Playground and Skyscraper reveal detectable Ca loss up to 15% in autumn and early winter. Frequently, higher PCP has been attributed to slower drip rate which allows the drip water to degas longer and precipitate while hanging on the cave ceiling (stalactite formation). However, in our drip water dataset, no relationship between the drip rate and the

Sr/Ca-index is found, and the months with the lowest Sr/Ca-index paradoxically happen to be months with increased drip rates (Fig. A1a-c). PCP also does not increase during negative P-PET so is not driven by an increase in air filled voids in karst system caused by the seasonal water deficit.

Potential stalagmite growth rate simulated with I-STAL using measured cave conditions suggests a seasonally biased growth in summer and winter with intervals of reduced or inhibited growth in spring and autumn (Fig. 6a). Growth inhibition in

autumn coincides with increased PCP as suggested by a decreased Sr/Ca-index (blue shading in Fig. 6). Stalagmite growth simulated in I-STAL can reach up to 100 µm yr$^{-1}$, and occasionally shows negative growth rates suggesting growth cessation (undersaturation). This results in a temporal hiatus in a stalagmite or even the dissolution of calcite. In the case of Playground, growth was only simulated for winter 2020/2021, as simulations of the previous winter suggest no stalagmite growth (Fig. 6a).





In bedrock sampled from fallen blocks throughout the cave, some elements exhibit a very high degree of spatial variation, at

times exceeding an order of magnitude (Table A2). To compare drip water and bedrock, we use the TE/Sr ratio which is relatively insensitive to the degree of PCP and/or additional sources (e.g. marine). Direct comparison of TE/Sr ratios of bedrock and drip water shows that some elements feature overlapping ranges in the bedrock and drip water (U, Ca, Li, Cu; Fig. 7). Other elements such as Na, Si and S have much higher ratios in drip water compared to bedrock, while Mg, Ba, As, K and Cr are moderately enriched. Elements such as Al, Y, and Mn have much lower ratios in drip water than in the sampled bedrock.

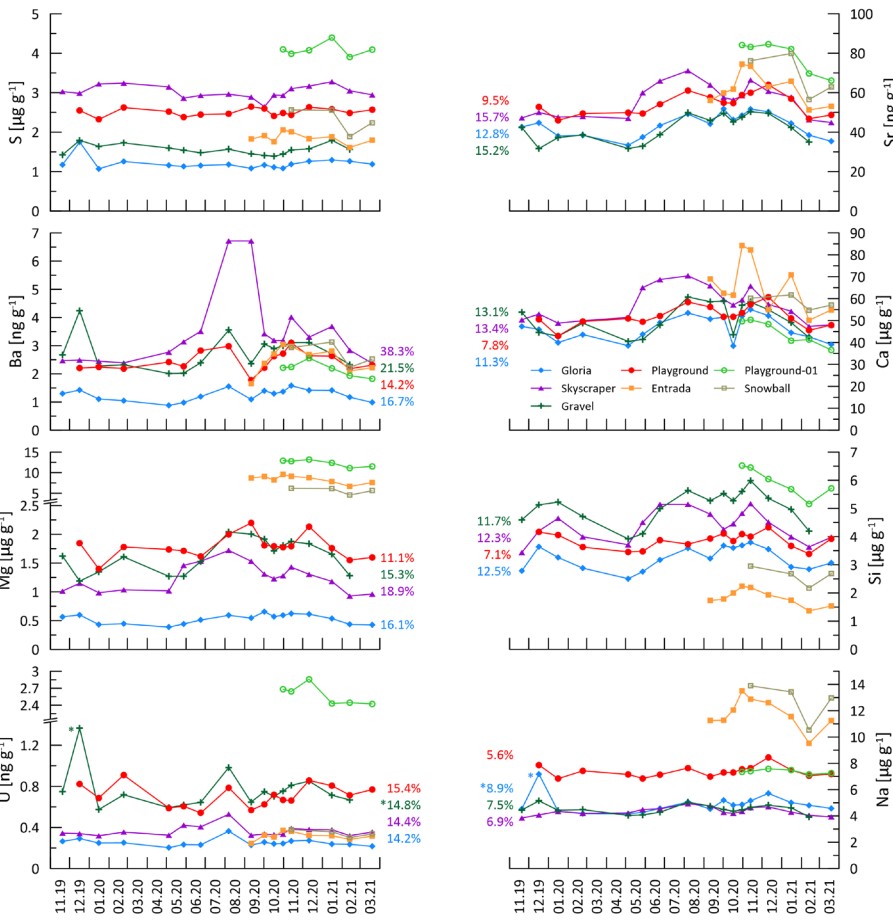


**Fig. 4: Analyzed elemental concentration. In the first part of this figure (PC 1 elements and sulfur) colored numbers give the %-RSD of variation to give an impression of how much the seasonal amplitude changes. Numbers with an asterisk are outlier cleared (outliers: >25% higher than 2ⁿᵈ highest concentration). The second part of the figure (next page) contains the spikier elements controlled by PC 2 and PC 3. Note that concentrations can be given in ppmV (µg g⁻¹) or in ppbV (ng g⁻¹). Missing data points denote**

**points below LOQ.**





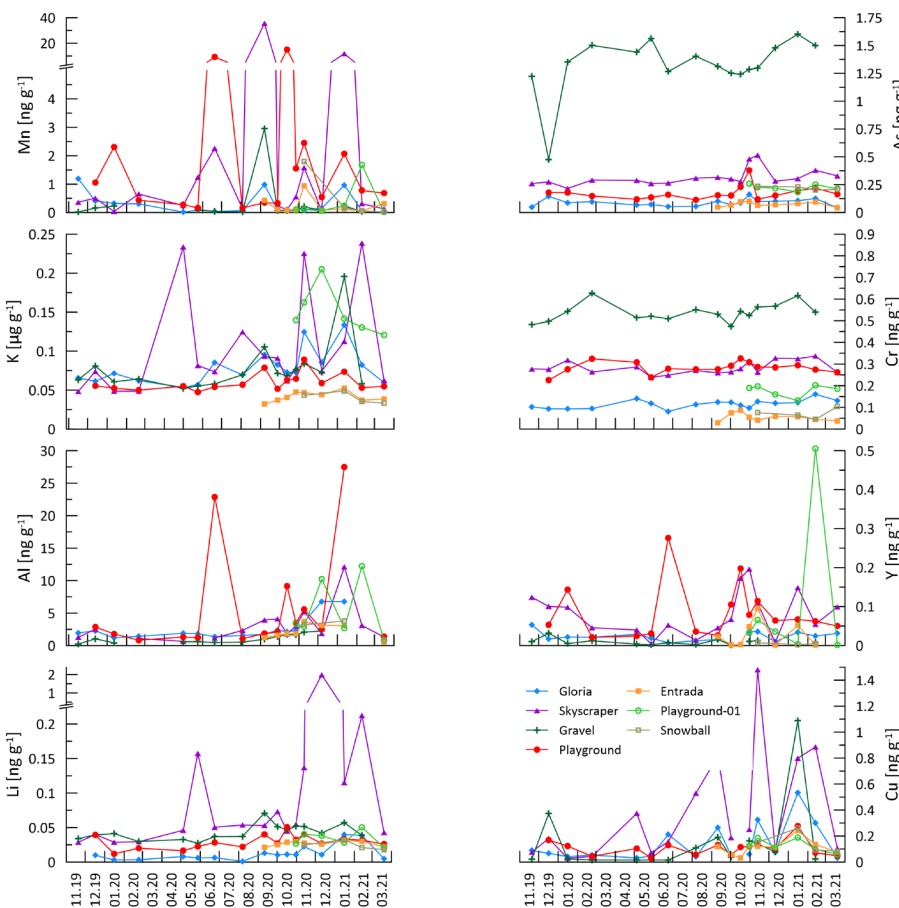

**Fig. 4 (continued)**

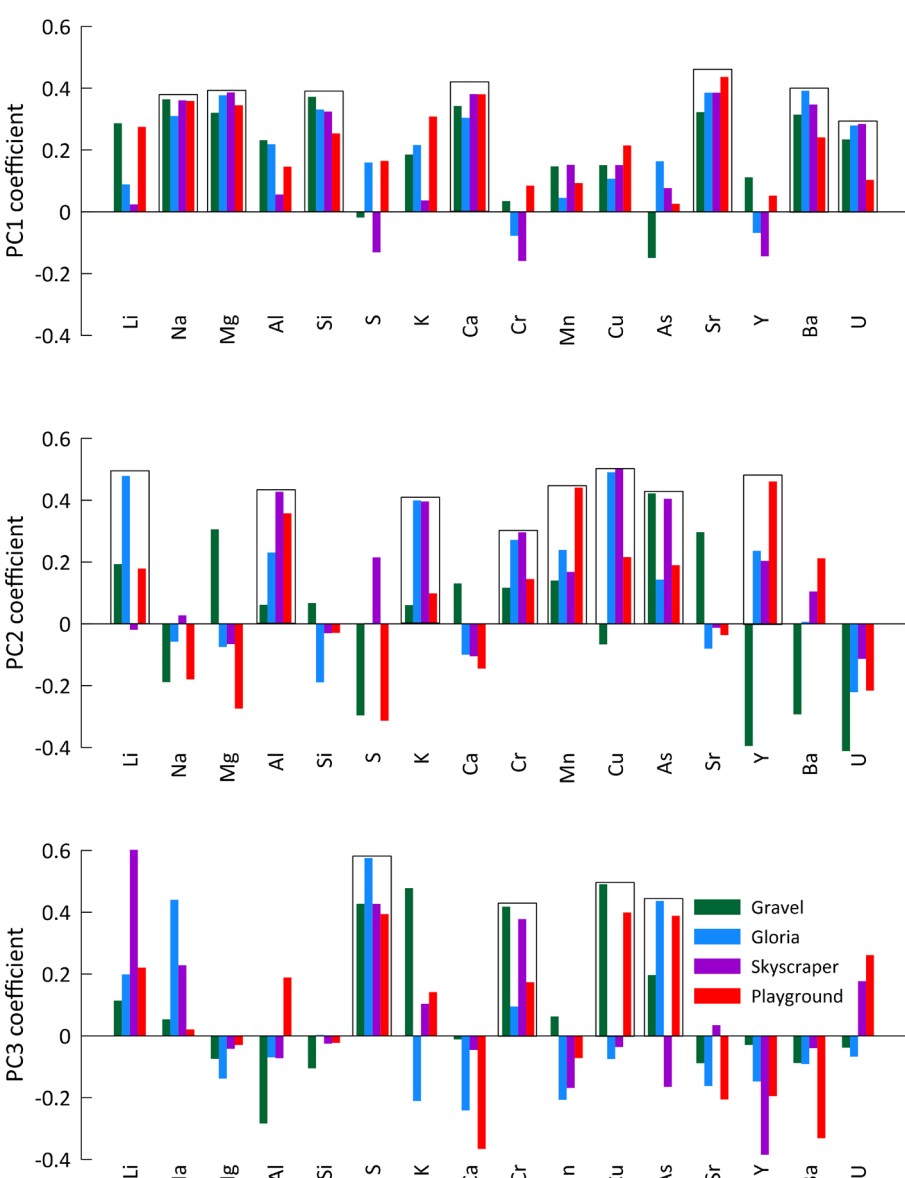

**Fig. 5: First three principal components (PC 1-3) of the principal component analysis (PCA) applied on the element concentration of drip water from individual drip sites. Boxes indicate elements of generally similar behavior controlled by specific PCs.**






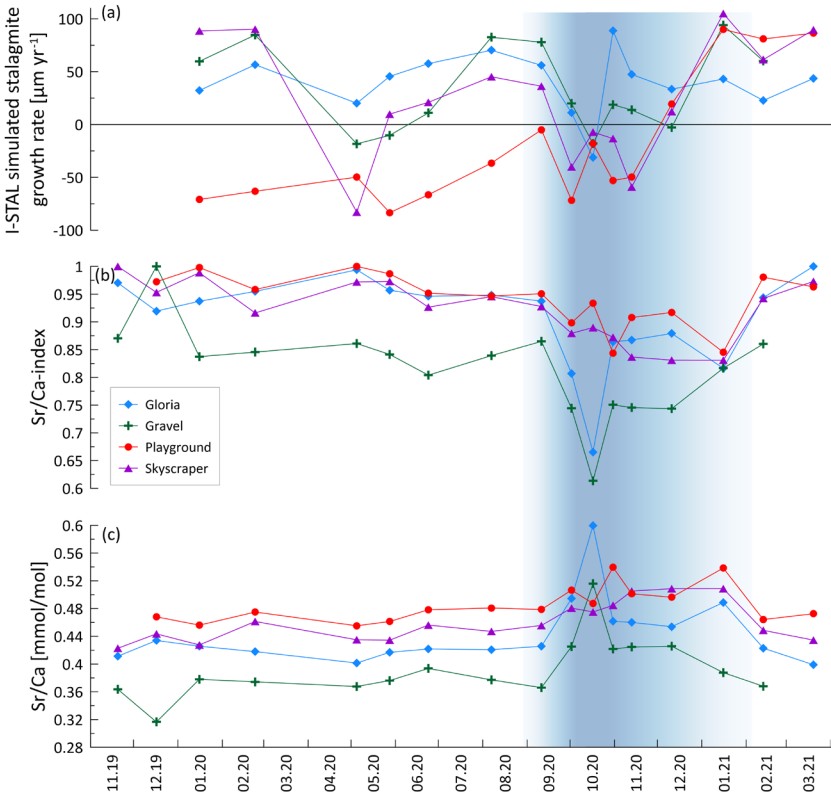

**Fig. 6: Prior Calcite Precipitation (PCP) indicated by Sr/Ca and its impact on growth rate. a) I-STAL simulated stalagmite growth rate, b) Sr/Ca- index and c) Sr/Ca ratio of measured drip water. The blue band indicates a phase of intensified PCP in some drip locations likely contributing to reduced stalagmite growth.**

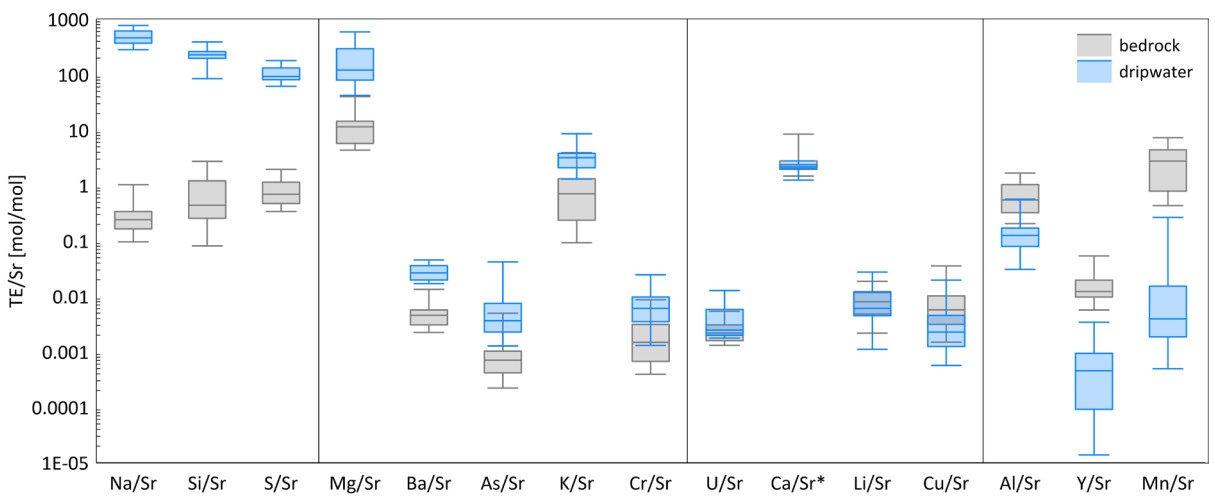

**Fig. 7: Box-and-Whisker diagram comparing measured ranges of bedrock and drip water TE/Sr ordered by difference of median between bedrock and drip water from left to right: enriched elements relative to bedrock (Na, Si and S), moderately enriched elements (Mg, Ba, As, K and Cr), congruent elements (U, Ca, Li and Cu) and depleted elements (Al, Y and Mn). \*Note that Ca/Sr is given in mol/mmol.**




## 5. Discussion

### 5.1 Seasonal evolution of cave air composition and estimation of δ¹³C of the respired endmember

The maximum cave air $pCO_2$ in summer and minimum in winter is consistent with seasonal ventilation of the cave controlled by the density contrast between cave and outside air, a process common to many caves in regions with significant seasonal temperature contrast (James et al., 2015). The cave $pCO_2$ approaches exterior values during periods when exterior temperature drops below cave temperature, triggering inflow of external air through the cave entrance. In warm periods, when outside air is warmer than the cave, advection through the entrance is limited, and the cave air composition approaches that of the overlying soil and epikarst. Although the monitoring resolution is low, it suggests that the ventilation regime responds quickly to exterior temperature, typically within a few days to a week. We also see evidence of variations in the efficiency of ventilation within the cave due to cave morphology. In the warm periods, high $CO_2$ preferentially accumulates in the lower, more isolated locations like Skyscraper and Playground. At the onset of ventilation (Nov-Dec 2020), decreasing $CO_2$ concentrations are first attained in these lower galleries (higher level locations remain high in $pCO_2$), where dense air inflow from the exterior concentrates in the lower portions of the cave. These differences within the cave affect the degree of drip water oversaturation and the seasonality of speleothem growth in different cave sectors (see section 5.4.2).

In addition to the ventilation effect, the $CO_2$ in the cave may also be affected by variations in the $CO_2$ concentration of the soil which may not always covary with exterior temperature. Although soil respiration rates generally are highest at higher temperatures, heterotrophic respiration also depends on the supply of labile organic matter and soil moisture (Krishna and Mohan, 2017). High cave $pCO_2$ concentrations in mid November 2020, despite exterior temperatures near the ventilation threshold, suggest a high $CO_2$ supply from the soil/epikarst which may reflect high rate of microbial degradation of recently fallen leaf litter as well as optimal moisture conditions and mild temperatures for soil respiration (Krishna and Mohan, 2017). Detailed soil air sampling studies have the potential to elucidate the seasonal concentration and isotopic variations of this endmember, but are beyond the scope of the present study.

As a result of the changing ventilation regime, $δ^{13}C_{air}$ of cave $CO_2$ varies along a mixing line defined by the exterior atmosphere and the respired $CO_2$ source in the soil and epikarst (Fig. 8). Using the measured forest values to define the local exterior endmember (-11.6‰), our data suggest an isotopic value for the respired endmember of -25.6‰, consistent with the range of recent C3 ecosystems (range: -24 to -32‰; e.g. Mattey et al., 2016; O'leary, 1988; Pataki et al., 2003). The large range of $pCO_2$ and $δ^{13}C_{air}$ values places tight constraints on the Keeling plot and the uncertainty of the intercept (SD of y-intercept = 0.21) is relatively small compared to other Keeling plot applications (Frisia et al., 2011; Pataki et al., 2003; Riechelmann et al., 2011). Unlike some other cave systems (Frisia et al., 2011; Waring et al., 2017), our cave air samples define a single line typical of a two component mixture, and do not show evidence of degassing of drip water as a significant influence on the isotopic ratio of cave $pCO_2$. That is, while the influence of drip water $CO_2$ degassing is likely low in summer due to low drip rates and high cave $pCO_2$, we do not see any evidence for a change in slope in winter due to higher drip rates and lower $pCO_2$.





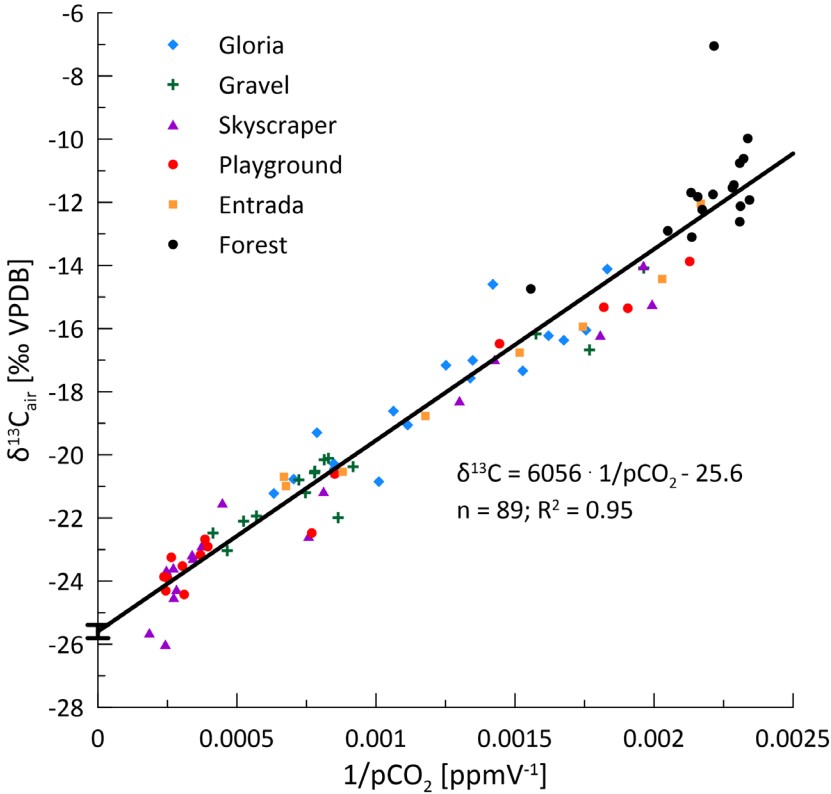

**Fig. 8: Keeling plot based on cave air measurements. The mixing line (black line) suggests a mixture between the two endmembers of respired soil air (-25.6‰) and atmospheric air (-11.6‰).**

**5.2 Meteoric water recharge and hydrology**

On a seasonal scale, drip rate is controlled by precipitation and potential evapotranspiration (P-PET), while the water reservoir feeding the drip sites recharges in the wet season (October-April) primarily by rainfall. In summer months, recharge is limited because rain events are sporadic and compensated by the higher evapotranspiration. The temporal point sampling resolution in this study makes it impossible to precisely quantify the lag behavior and response on rain events affecting the drip rates at

monitored sites. Continuous hydrographs recording drip rates or tracer experiments would be necessary to determine the response time of rain events and to define thresholds of rainfall amount affecting drip sites individually (Markowska et al., 2016). The proposed 7-day antecedent cumulative rainfall method (Baker et al., 2020; Baker et al., 2021) does not yield any better relationship between recharge and drip rate. Nonlinear relationship between rain and drip rate can result from passing the precipitation vs. evapotranspiration threshold (Fairchild and Baker, 2012) or when water storage reservoirs overflow

(Bradley et al., 2010). Nevertheless, some site specific response behaviors at the onset of the rainy season are observed. Despite thickest overlying bedrock, Skyscraper and Playground respond quickly to the onset of the rainy season (likely related to the "piston effect" pushing water through the reservoir), while Gravel and Entrada react in a delayed fashion. Drip rates of Entrada and Playground mirror the structure of the monthly recharge (P-PET) suggesting a close relationship with potential recharge





(Fig. 3). Despite similar total rainfall amounts during the two monitored wet seasons, drip rate differences are observed between
drip sites (e.g. Gloria). This might be associated with the hydrological history or size of the discrete stores. Alternatively, a
change in flow path routing could explain this (Markowska et al., 2016). For example, Gloria has higher drip rates during the
2019-2020 wet season than the 2020-2021, which could reflect higher evapotranspiration during the warmer 2020-2021 winter.
However, Gravel exhibits the opposite behavior which may reflect complexities in hydrological routing at this site such as
overflow reservoirs (Bradley et al., 2010) which are consistent with the variation of which soda straw hosts drip at this location.
The $\delta D$ and $\delta^{18}O$ of drip waters provide some further constraint on the hydrological behavior. We describe first the behavior
of individual drips. The several permil variations between amount weighted $\delta^{18}O_{rain}$ of successive months in some years
(Moreno et al., 2021) is not manifest as seasonal cycles in $\delta^{18}O_{dripwater}$ (Fig. 3 and Fig. A2). No change in the drip water isotopic
composition over the onset of the wet season is observed. The majority of our sampled drip locations feature very limited
(<1.1‰) range in $\delta^{18}O$ across the monitored period, although precipitation events span over 10‰ (Fig. 9). Within a given drip
location, the narrow $\delta^{18}O$ suggests mixing of individual recharge events within the flow and storage system of that drip, so that
the isotopic signatures of individual precipitation events are attenuated. Similar attenuation of event scale variability has been
described previously (e.g. Riechelmann et al., 2011; Feng et al., 2014; Tadros et al., 2016). In a region with similar widespread
range in $\delta^{18}O$ of precipitation events, such attenuation of annual range has classified drips with 1‰ variability as well-mixed
cave waters characterizing diffuse flows (Domínguez-Villar et al., 2021). In our monitoring, the drip site exhibiting the largest
range (1.9‰) was Entrada. Such a level of variance (e.g. up to 3‰) has been described to result from partially-well mixed
waters which are fed by preferential drainage routes (Domínguez-Villar et al., 2021). We note that Entrada features a small
bedrock thickness (7m), and also large trees with root penetration into the cave. Root channels have been observed to provide
preferential flow routes in some karst systems (Dasgupta et al., 2006). The Gravel location also features slightly greater (1.4‰)
variation than other locations and also is near root systems and is inferred to reflect a sediment filled chimney which may also
have altered flow pathways compared to limestone. Despite the overall high attenuation of individual events, at the end of the
monitoring, $\delta D$ and $\delta^{18}O$ at all drip sites (except Playground) shifts negative, suggesting a change in the water composition of
those discrete reservoirs. We hypothesize a rain event with strongly negative isotopic compositions in early February to cause
such a shift. Strongly negative rainfall events of substantial volume are possible, as seen in Fig. 9, which could recharge the
reservoir during conditions with high recharge.
The spatial variations in average $\delta^{18}O$ of drip waters suggest that there is not a single well-mixed reservoir that supplies all
drip locations throughout the cave. The spatial differences may reflect variation in the dominance of diffuse and preferential
flow routes and differing transit times. From our dataset, we are unable to quantify the water transit time, because rain $\delta^{18}O$
monitoring does not cover the year of drip sampling nor the prior year. Thus, we cannot rule out the possibility that inter site
differences may reflect varying average transit times among drips, so that the amount weighted drip water $\delta^{18}O$ of different
drips would converge over a suitably long time period which averaged interannual differences in amount weighted $\delta^{18}O$ of
precipitation. If significant differences in mean transit times existed among drip sites, and the amount weighted average
precipitation in the year prior to monitoring were more positive than that during the monitored season, then drips with slower




average transit times might be offset to the previous year's value. Interannual variation of about 1.25‰ in the yearly precipitation amount weighted average $\delta^{18}O$ is seen among the available precipitation data (Fig. 9) which includes 15 years

from GNIP station Santander in the period 2000-2015 (no recent data available; Iaea/Wmo, 2022) and individual rain events monitored near Oviedo from February 2015 to February 2016 (Moreno et al., 2021). In the 2015-2016 series, considering the amount weighted average of months with P-PET > 0 (months with recharge; brown cross), the $\delta^{18}O$ is 0.3‰ more negative compared to the full annual rain (Fig. 9; green cross) (Moreno et al., 2021). We observe that at drip sites located under the same bedrock thickness and vegetation, such as Playground and Playground-01 (located 3 m apart in the same gallery), the

slower drip rate (Playground-01) features a more positive average $\delta^{18}O$ and $\delta D$, which could be compatible with a slower average transit time.

Additionally, the flow pathways can also affect which precipitation events recharge the aquifer. In some settings dominated by diffuse recharge through the soil, summer rainfall might not contribute to the water reservoir if the recharge threshold is not exceeded. However, this effect is suggested to be minor in caves with P/PET higher than 1.5 (Baker et al., 2019). In our

setting (P/PET 2.4), the amount weighted $\delta^{18}O$ during months of positive water balance is only 0.3 ‰ lower in $\delta^{18}O$. Nonetheless, preferential flow routes may allow recharge even in months of negative water balance, which in some sites may result in recharge from the less negative precipitation events which characterize the summer months (Domínguez-Villar et al., 2021). Sites like Entrada with proximity to root systems entering the cave and facilitating preferential flow, feature a higher average $\delta^{18}O$ and $\delta D$ than sites with comparable or greater drip rate but no large trees or roots, like Gloria or Playground. The

spatially variable vegetation, with variable transpiration water demand, may also affect the recharge threshold. Finally, isotopic fractionation of rainwater by evaporation in the soil and the unsaturated zone before entering the discrete water reservoir could contribute to the slight shallowing of the drip water meteoric water line (DMWL) relative to the local meteoric water line (LMWL), and variable influence of evaporation could contribute to the spatial pattern in mean drip water ($\delta^{18}O$), similar to Baker et al. (2019) and Bradley et al. (2010).



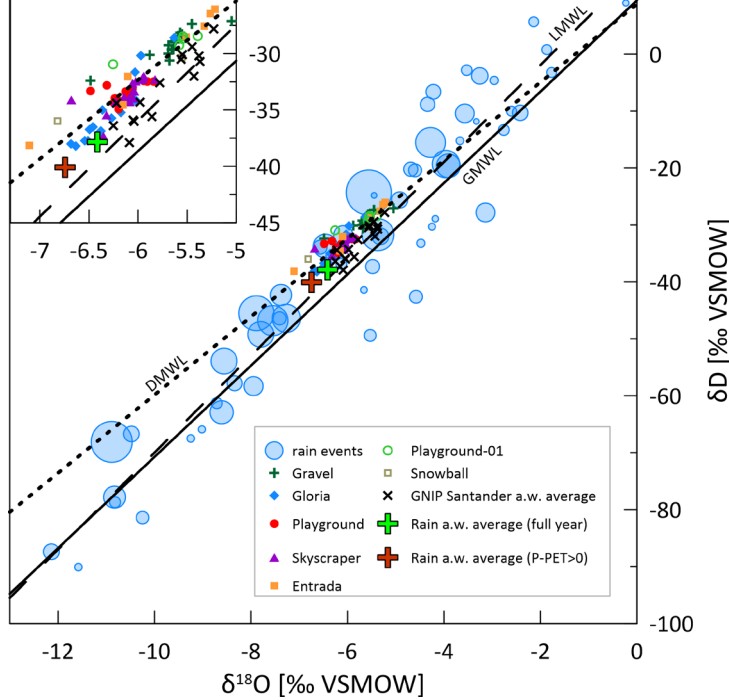


**Fig. 9: Meteoric water lines of drip water and local rain events. The size of the blue bubbles indicates the rain amount of individual rain events between February 2015 to February 2016 (Moreno et al., 2021). The green cross represents the full annual amount weighted average of local precipitation. The brown cross accounts for positive P-PET months only (recharge). Black crosses reveal the interannual variability of rainfall represented by amount weighted annual average of GNIP station Santander for 2000-2015.**
**For the 2019-2021 cave monitoring, drip water DMWL: δD = 7.01 δ$^{18}$O + 9.72, where R$^2$=0.83; stippled. Local rainwater LMWL δD = 8.5 δ$^{18}$O + 15.1, where R$^2$=0.92; dashed (data from Moreno et al. (2021)) and global meteoric water line (GMWL) δD = 8.0 δ$^{18}$O + 10 (Craig, 1961). The MWL for GNIP Santander 2000-2015 (not shown) is fairly similar to the dashed line.**

These findings of a moderately- to well-mixed water reservoir, likely recording a mean annual amount weighted δ$^{18}$O signal and buffered against individual rain water recharge events (especially during the dry season when (P-PET) is negative; Fig.
3), is crucial when interpreting stalagmite δ$^{18}$O records. Site specific evaporative effects on δ$^{18}$O$_{dripwater}$ can lead to offsets in coeval speleothems from the same cave. These effects may be more extreme now due to spatially variable vegetation and extreme evapotranspiration of perennially green Eucalyptus, compared to native deciduous oaks. Nonetheless, sites of slow drip rate could in some cases be characterized by more positive δ$^{18}$O of drip water and therefore calcite, than sites of more rapid drip flow due to stronger or weaker evaporative effects along the drip water flow path respectively. This may be consistent
with observations of slowly growing stalagmites having higher δ$^{18}$O (Stoll et al., 2015), if slow growth were in part caused by slow drip rates and potentially concomitant high levels of PCP. Thus, disequilibrium effects in δ$^{18}$O incorporation into speleothem calcite may not be required to explain differences in δ$^{18}$O in coeval speleothems in the same cave. Stalagmites from cave sectors with thinner overlying bedrock thickness may result in a less well-mixed water reservoir and thus reflect seasonal changes in rainwater δ$^{18}$O recharge.





### 5.3 Controls of $\delta^{13}C_{DIC}$

Several factors may potentially contribute to the strong seasonality in drip water $\delta^{13}C_{DIC}$ marked by higher $\delta^{13}C_{DIC}$ in the winter season. A decrease in soil $pCO_2$ in colder seasons could produce this trend because the $\delta^{13}C$ of soil $CO_2$ is expected to fall on the mixing line between atmosphere and a respired endmember (similar to Fig. 8), so that periods of higher soil $pCO_2$ are characterized by lower $\delta^{13}C$ of the soil $CO_2$ and therefore the DIC in drip waters. Although we have no direct soil $CO_2$ measurements, soil $pCO_2$ in similar climates is elevated in the warm season, and can be enhanced by greater soil moisture (Borsato et al., 2015; Frisia et al., 2011; Hasenmueller et al., 2015; Hodges et al., 2019). A shift to more closed system dissolution in the colder and wetter season could also contribute to the trend of higher $\delta^{13}C_{DIC}$ in the colder seasons. Additionally, equilibration of the drip water with cave air could also lead to a seasonal trend because of the higher $\delta^{13}C_{air}$ of cave air in the ventilated winter season. Finally, if PCP were more extensive in the winter season it would also increase $\delta^{13}C_{DIC}$ due to preferential degassing of light $^{12}C$ carbon isotopes when cave air $pCO_2$ is low. To evaluate these processes, we consider the indicators of PCP and degree of bedrock dissolution as well as calculations generated by the CaveCalc model (Owen et al., 2018).

Our data on PCP suggest that PCP cannot be the main driver of seasonal variations in the $\delta^{13}C_{DIC}$ in any of our three sites with full 16 month monitoring data. The range in PCP in most sites is small and is modeled to have only a limited effect on the $\delta^{13}C_{DIC}$, and the samples of high $\delta^{13}C_{DIC}$ do not correspond to the samples of highest PCP (Fig. 10). Variations in soil $pCO_2$ with concomitant variations in $\delta^{13}C$ of soil $CO_2$ defined by a mixing line between atmospheric $CO_2$ and respired $CO_2$ (Fig. 8) could explain the observed covariation of Ca and $\delta^{13}C_{DIC}$ in some sites (e.g. Gloria), which reflects the greater degree of bedrock dissolution accompanying higher soil $pCO_2$ (increased Ca concentration). However, the very large range of $\delta^{13}C_{DIC}$ at constant Ca and PCP, especially in Playground (and Gravel) is not consistent with variation in soil $pCO_2$ as the only process.

Previous speleothem based studies of the $^{14}C$ dead carbon fraction (DCF) in stalagmites in La Vallina and nearby caves show average DCF values from 3-12% (Lechleitner et al., 2021) which suggests mostly open system dissolution. Nonetheless, a shift from very open (3% DCF) to moderately open (12% DCF) would be accompanied by a 0.75‰ increase in the $\delta^{13}C_{DIC}$, which is minor compared to the seasonal variations of several permille (Fig. 2). In tropical settings, DCF has been shown to increase with higher water flow (Noronha et al., 2014). Thus, periods of higher drip rates in the cold season could, in principle, lead to lower gas exchange volumes and more closed system dissolution, contributing to the seasonal increase in $\delta^{13}C_{DIC}$. The similarity of $F^{14}C$ in drip water of multiple seasons, equivalent to a DCF of 5-8%, if respired carbon has $F^{14}C=1$, suggests that seasonal variation in DCF is not large enough to explain the large observed shift in $\delta^{13}C_{DIC}$ (Fig. A3).

Two additional processes not explorable with these CaveCalc simulations are (1) a decrease in the $\delta^{13}C$ of the respired endmember in soils, and (2) kinetically enhanced degassing and exchange with cave air. A change in the $\delta^{13}C$ of the respired endmember, driving a change in $\delta^{13}C_{DIC}$ without concomitant change in Ca, is difficult to detect in our cave air monitoring because the cave air in the coldest times samples ventilated atmosphere air not soil and epikarst air. Such a change could occur if there were a change in the relative importance of root vs heterotrophic respiration or the importance of soil vs bedrock





organic matter (OM) sources if these featured OM of contrasting $\delta^{13}C$. We find strong seasonal variations at Playground (8.6‰ $\delta^{13}C_{DIC}$ range) which is covered by pasture, whereas other monitoring sites are located beneath bushes or trees and undergo

less variability in $\delta^{13}C_{DIC}$. Deeper roots of larger plants could maintain higher autotrophic respiration in winter in these sites, compared to shallow rooted grasses, especially since the temperature range in the surface soil occupied by grass roots is much larger than that at depths of a few meters spanned by tree roots (Benzaama et al., 2018). Local changes in $\delta^{13}C$ of soil $CO_2$ could further be controlled by vegetation cover since microbial degradation of fallen leaves in autumn and winter might contribute substantially to the negative $\delta^{13}C_{DIC}$ imprint in late autumn.

Kinetically enhanced degassing and exchange with cave air potentially influences the $\delta^{13}C_{DIC}$ at some drip sites (e.g. Playground or Gravel) since PCP and DCF alone did not explain the variations in $\delta^{13}C_{DIC}$. Here, $\delta^{13}C_{DIC}$ is hypothesized to be affected not only by the source $\delta^{13}C$ of the soil, but also the degree of drip water degassing (i.e. increased degassing leads to more positive $\delta^{13}C_{DIC}$ due to preferential loss of $^{12}C$). A cave experiment where drip water was sampled along a flowstone to track the degassing and $CaCO_3$ precipitation evolution likely due to cave air exchange results in a steeper slope (blue line in

Fig. 10) compared to the CaveCalc modeled degassing line (grey dashed line in Fig. 10). This suggests that kinetically enhanced exchange with cave air might be important since $\delta^{13}C_{DIC}$ changes by almost 6‰ without losing appreciable Ca and comparable Sr/Ca-index ranges (numbers along blue line) compared to the CaveCalc degassing line (dashed). For Gravel and Playground, a statistically significant correlation between cave air $\delta^{13}C_{air}$ and drip water $\delta^{13}C_{DIC}$ emerges suggesting exchange with cave air (Fig. A4). Such a correlation with cave air $\delta^{13}C$ could be a causal mechanism (equilibration) or an accidental

correlation because cave air $\delta^{13}C$ and drip water $\delta^{13}C_{DIC}$ are both controlled by temperature. The slope of the in-cave derived degassing line (blue line) appears parallel to the drip water samples (e.g. Playground) with least negative $\delta^{13}C_{DIC}$ values and no substantial loss in Ca, and therefore high Sr/Ca-index.

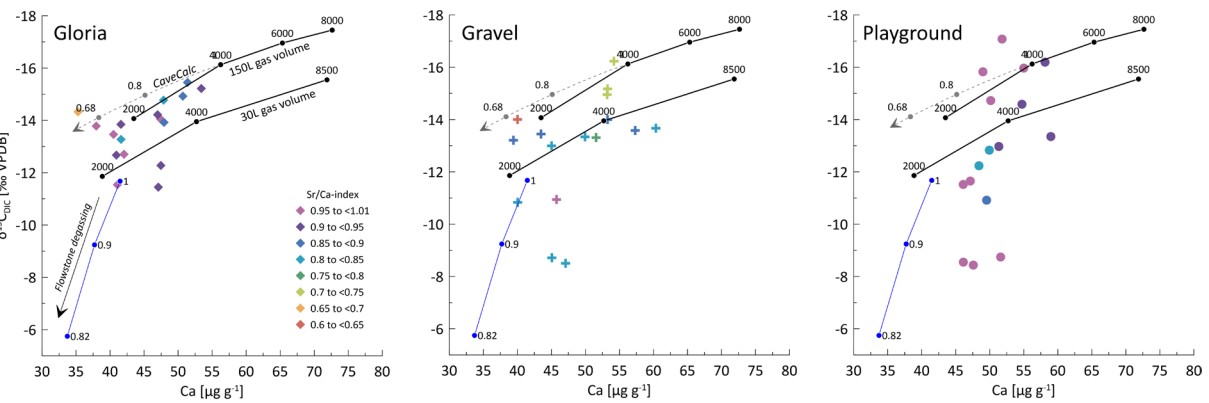

**Fig. 10: CaveCalc simulations compared with monitoring data of three individual sites (Gloria, Gravel and Playground) color coded**
**by Sr/Ca-index ranges. Black markings show CaveCalc simulations (Owen et al., 2018) with different gas volume (150L and 30L) representing more open and closed system, respectively. Along the black line the soil $pCO_2$ (labels) is changing using a respired endmember $\delta^{13}C$ found by the Keeling plot (Fig. 8). The dashed line projects the modeled evolution of degassing starting with a 4,000 ppmV soil $pCO_2$ labeled with the fraction of Ca remaining (Sr/Ca-index). The blue line marks a degassing evolution along a flowstone measured during the monitoring campaign, where labels indicate remaining Ca fraction (Sr/Ca-index).**



These results indicate strong variations in $\delta^{13}C_{DIC}$ which reflect both a temperature control on biomass (respired endmember) and potential controls by kinetic degassing (PCP) or equilibration with cave air. No evidence for a substantial dead carbon contribution from dissolved bedrock is found (Fig. A3). Our results are consistent with the long-term suggestion of strong temperature control of speleothem $\delta^{13}C$ (e.g. Genty et al., 2006; Lechleitner et al., 2021; Moreno et al., 2010). Significant seasonality in drip water $\delta^{13}C_{DIC}$ means that interpretation of speleothem $\delta^{13}C$ may need to account for seasonality of calcite

deposition where calcite deposition is not year-round (further discussion in section 5.4.2).

## 5.4 Seasonal variation in bedrock dissolution and saturation state

### 5.4.1 Seasonal cycle in bedrock dissolution

Elements dissolved congruently from limestone, including Ca and Sr (and other PC 1 elements), follow a seasonal cycle with a similar amplitude as shown in Fig. 4 (indicated by the %-RSD of element concentration). The cycle is interpreted to result

from changing bedrock dissolution intensity (colored lines in Fig. 11) driven by temperature (shaded range in Fig. 11) with higher dissolution in summer and autumn months due to higher soil $CO_2$ concentration and lower in winter and early spring (represented by dashed line in Fig. 11). While PCP could in principle affect the concentration of Ca, it has a very limited effect on the concentration of elements with partitioning coefficients much lower than 1, as is the case for Sr (Wassenburg et al., 2020) and U (Weremeichik et al., 2017). Therefore, the similar amplitude variation of Ca, Sr, and U suggests that the seasonal

cycle in Ca concentration is also predominantly controlled by bedrock dissolution intensity. Seasonal summer and autumn maxima in soil $pCO_2$ was interpreted from changes in $\delta^{13}C_{DIC}$ (dashed line in Fig. 11), which would result in stronger bedrock dissolution because of the higher $CO_2$ acidifying the percolating water. The inverse relationship between Sr concentration and $\delta^{13}C_{DIC}$ supports this finding (Fig. A5). The effect of higher soil $pCO_2$ may also be amplified by more open system dissolution conditions which would enhance the degree of bedrock dissolution as $CO_2$ is continuously supplied and maintains the acidity

of percolating waters.

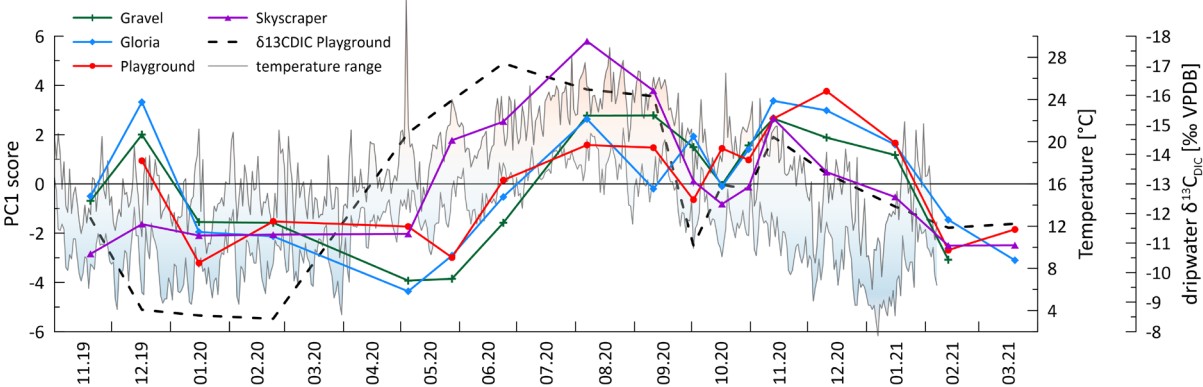

**Fig. 11: The temporal evolution of the score of PC 1 (colored curves) suggests stronger bedrock dissolution in summer and early autumn as well as winter likely due to higher soil pCO₂. A clear relationship with temperature (daily min-max with color shading) and $\delta^{13}C_{DIC}$ (dashed line, note inverted y-scale) emerges suggesting an interrelationship.**



### 5.4.2 Drivers of temporal variations in PCP and growth rate


PCP is expected to be maximized by slow drip rates (long exposure), high drip water oversaturation due to high initial Ca from bedrock dissolution and low cave $pCO_2$ from cold season ventilation. Because the former factor is maximized in summer season, but the ventilation works to increase the latter effect in the cold season, PCP shows little seasonal variation in many sites (e.g. Playground and Skyscraper; Fig. 6). At sites with pronounced variation in PCP in October 2020 (Gravel and Gloria), the minimum in the Sr/Ca-index occurs when drip rate just starts increasing (Fig. A1c does not indicate strong correlation). Hardly any Ca loss would be expected for rapid flow rate of drip site Skyscraper, which limits the timescale of degassing as the water is replaced and mixed immediately with calcite oversaturated drip water. However, a substantial loss of 17% is observed at Skyscraper. One explanation may be that the drip rate results from convergent flow and that the opportunity for PCP occurred at higher levels in the karst system. Alternatively, in other settings, PCP seasonality has been attributed to seasonal cave ventilation cycles which increase drip water oversaturation (Mattey et al., 2016). In La Vallina, the cave starts to ventilate around the time when the Sr/Ca-index indicates increased PCP (Figs. 2 & 6). Consequently, the stronger $CO_2$ gradient between enriched drip water and ventilated cave air enhances degassing and favors PCP. Direct comparison of the drip site specific Sr/Ca-index with the local cave air $pCO_2$ does not show a strong relationship (Fig. A1b). However, Gravel and Gloria indicate substantial Ca loss in October likely related to a strong $CO_2$ gradient due to a ventilation event while Skyscraper and Playground do not show such a significant PCP imprint because the cave air $CO_2$ is still relatively high deeper in the cave. The $CO_2$ gradient can also be strengthened by a higher drip water carbonate concentration when soil $pCO_2$ increases. Indeed, stronger bedrock dissolution represented by higher Sr concentration anticorrelates with Sr/Ca-index suggesting increased PCP with higher soil $pCO_2$ (Fig. A1a). Non-linear relationships of driving parameters likely complicate this interpretation. Nevertheless, a seasonal maximum in the extent of PCP is likely conditioned by multiple variables including low cave $pCO_2$, high soil $pCO_2$, and potentially low drip rate. We suggest that the seasonal PCP regulation in La Vallina is complex because the effect of lower summer drip rate is counteracted by the effect of higher summer cave $pCO_2$. Systems with coincident low summer drip rate and low summer cave $pCO_2$, such as in Gibraltar (Mattey et al., 2016), have shown a strong seasonal control of PCP. Further modeling and proxy data analysis could constrain the mechanisms controlling PCP.

It is crucial to understand under what conditions and season stalagmites effectively grow when interpreting their proxy signals. Using monitoring data as input parameters, stalagmite growth can be simulated with I-STAL (Stoll et al., 2012). A seasonal structure with growth phases in summer and winter interrupted by growth cessations in spring and autumn driven by cave conditions (cave air and hydrology), cave processes (PCP) as well as drip water geochemistry (carbonate saturation) emerges (Fig. 6). The simulated growth rates match observations from La Vallina cave with annual growth rates ranging from a few micrometers up to 100 $\mu$m yr$^{-1}$ (Stoll et al., 2015; Stoll et al., 2013). Future studies comparing simulated stalagmite growth with natural samples formed beneath the same drip would provide an opportunity to validate model output. Modeling results demonstrate that there is a complex interplay between Ca concentration and $CO_2$ in drip water relative to the cave environment ($pCO_2$ and drip interval) at a constant temperature, as well as increased PCP inhibiting stalagmite growth in autumn. In the





case of Gravel, Skyscraper and Gloria, reduction of drip water supersaturation due to PCP is sufficient to cause growth cessation in this season. Nevertheless, lower cave air $pCO_2$ in these two locations leads to pronounced growth rates in these

sites during winter months. An exception is Playground, which is simulated to only grow during the last few months of the monitoring period. Stalagmites growing all season with only short phases of inhibited growth (e.g. Gloria) might exhibit different geochemical signals than seasonally biased stalagmites (e.g. Playground only growing in winter), which should be accounted for when interpreting analyses derived from micromilling techniques which most often do not resolve subannual changes but are averaging longer periods (several years). Furthermore, periods without growth or with reduced growth can

alter the seasonal pattern of TE (Borsato et al., 2007) or subannual stable isotopes (Johnson et al., 2006), and therefore impact the interpretation of stalagmite geochemical analysis from microscale measurements (e.g. seasonal layers in fluorescence images or LA-ICP-MS tracks).

### 5.4.3 Other sources for enriched elements compared to bedrock

Elements with high PC 1 loading coefficients (Na, Si, Mg and Ba) show a similar seasonal pattern as Ca, Sr, and U which were

primarily controlled by bedrock dissolution (section 5.4.1). However, drip water ratios of Na/Sr, Si/Sr, Mg/Sr, and Ba/Sr are significantly higher than those of the measured bedrock samples (Fig. 7), suggesting these elements have additional sources beyond congruent dissolution of the limestone bedrock which we sampled or incongruent dissolution takes place.

The temporal variations in drip water Na concentrations are only about 50 to 80% of the amplitude of those of Ca (%-RSD in Fig. 4), an attenuation which is consistent with dual sources of Na to drip waters. That is, if limestone dissolution is seasonally

variable but the non-limestone Na source were constant or its delivery buffered by cation exchange in soils (Tadros et al., 2019), this would lead to attenuation of the seasonal pattern of variation in the limestone sourced Na. One likely contributor to Na in drip water is wet and dry deposition of marine aerosols on the land surface above the cave, due to the proximity of the cave to the Atlantic Ocean coastline (Tremaine et al., 2016). No clear relationship is found between Na concentration and proximity of trees (which would act as aerosol traps). Pastures capture marine aerosols less efficiently, however, at monitoring

sites beneath pasture Na concentrations are elevated (Playground and Playground-01). This might be related to salt added to animal feed or excrements of animals containing salts (Hamamoto and Uchida, 2015). Direct advection of marine aerosols into the cave by aeolian input may be significant near the cave entrance where Na concentrations are generally higher (Entrada, Snowball; Dredge et al., 2013). Since Na is dominated by marine and potentially anthropogenic sources, no evidence for cation exchange with clay minerals in the soil or karst acting as hydrologically controlled sink/source is found as observed in clay

rich settings such as in SE Australia (Tadros et al., 2019).

Assuming marine aerosol contribution for Na, similar trends should be visible in Mg and S as well. The inferred direct marine aerosol deposition near the entrance at Entrada could contribute to the correlation of Mg/Sr with Na/Sr which is only observed at Entrada (Fig. A6). Fittingly, this site also has slightly higher variability in Mg/Sr at Entrada (9.0% RSD; Fig. 12). However, marine contribution alone cannot be the only cause for the elevated Mg and S concentrations at the different drip sites. Notably,

Playground-01, Entrada and Snowball have substantially higher Mg concentrations decoupled from Na (Fig. 4). We suggest



that variations in the Mg (and S) concentration in different bedrock zones drives this variation. The presence of minor dolomite phases in some sectors of the host rock would be an important source of higher Mg (Tremaine and Froelich, 2013). Our analyzed bedrock samples do not yield the high Mg concentrations expected from pure dolomite. However, our collection may not sample the full range of bedrock compositions, and bedrock exposures in some locations show textures typical of
weathering residue of dolomite. Within the flow route of a given drip water site, lower solubility of dolomitic components of host rock can lead to higher drip water Mg content with longer water-rock interaction times (Fairchild and Baker, 2012). At all monitored sites, the Mg/Sr ratio of drip water varies by less than 10% despite order of magnitude changes in drip rate (Fig. 12). This suggests constant background Mg contribution and congruent bedrock dissolution over the monitored timescale. Nonetheless, changes in flow paths over long time scales could potentially cause infiltrating water to interact with bedrock of
a different composition, a factor which may need to be considered in interpretation of long-term stalagmite trace element records.

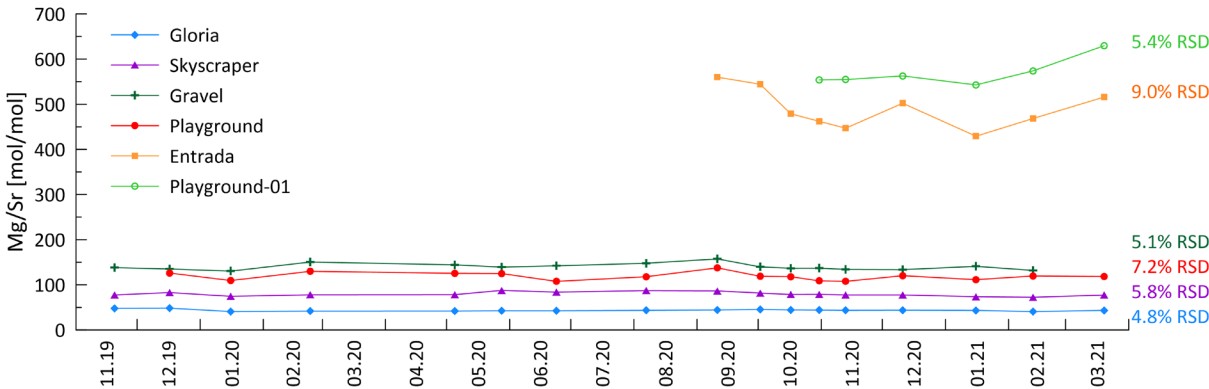

**Fig. 12: Constant Mg/Sr ratio over monitoring period indicates congruent bedrock dissolution. Higher RSD is recorded at Entrada (orange) which might be influenced by aeolian marine aerosols input near the cave entrance (Fig. A6). Mg contributed from**
**dolomitized zones might contribute to substantially higher Mg/Sr at sites like Entrada and Playground-01.**

Elements like Si and Ba, which are also observed to be substantially enriched relative to congruent dissolution of sampled bedrock, might be enhanced by leaching from silicate phases such as quartz rich sands which are present as infills in both levels of the cave or from clay minerals (Eylem et al., 1990). Ba can also be released by desorption from hydrous ferric oxides when pH is changing (Mishra and Tiwary, 1999). Calculations performed with PHREEQC (Parkhurst and Appelo, 2013)
suggest that at cave conditions the highest measured drip water Si concentrations are close to saturation relative to quartz, which may explain the low amplitude variations of the concentrations (Fig. 4). Longer water-rock contact times in summer may explain the slight increase in concentrations in summer.

**5.5 Episodic delivery of colloidal elements**

The seasonal scale variation observed in the major cations does not apply universally. Transition metals, Al, Y as well as Li
and K, show different patterns of variation, and are most strongly represented by PC 2 (PC 2 elements: Li, Al, K, Cr, Mn, Cu, As and Y; Fig. 5). Of these, Li, Al, K, Mn, Cu, and Y feature high amplitude short term variations, at least in some drip sites





(Fig. 4). Insoluble elements such as Al, Mn, Cu and Y have previously been attributed to detrital and colloidal phases, rather than dissolved transport, with colloids strongly associated with organic macromolecules (e.g. Hartland et al., 2012; Hartland et al., 2014). The variations reflected in PC 2 indicate that high concentrations of these elements occur predominantly in the

wet season (Fig. A7) although no relationships between hydrology and any elements emerges. As discussed earlier (section 5.2) the sampling resolution is poorly suited to test relationships between rainfall and drip rate, and therefore complicates interpretation of hydrological control on drip water chemistry. Aggregate monthly collections between sampling days would potentially be necessary to show better correlations between recharge and colloidally delivered elements. Since the present study targeted to acquire high fidelity geochemical data (as opposed to aggregate drip water chemistry), the protocol is not set

up to test this.

Some transition metals (Cu, Cr and As) which might be expected to be colloidally transported exhibit a temporal variation different from Al, Mn, and Y (PC 3; Fig. A7), suggesting additional controlling processes. For instance, ligand binding efficiency may be different for some of these elements or they may be in different sizes of colloidal fractions (Hartland et al., 2012; Hartland et al., 2014). Biomass take up and release could be another possibility complicating the interpretation of these

elements. Sulfur is dominantly controlled by PC 3 and is related to Cu, Cr and As (Fig. 5). Sulfur could potentially be affected by variations in bedrock pyrite oxidation. Other possibilities include redox changes and oxygen exposure related to the hydrology and contact with reducing materials (affecting the speciation of some of these elements) or the release of biologically bound elements when vegetation degrades at the end of the growing season (e.g. As, Cu, Cr; Kabata-Pendias and Pendias, 2000).

In many stalagmites, elements like Cr, Mn, Cu or Y covary seasonally (Borsato et al., 2007; Sliwinski and Stoll, 2021). Similar observations were made in drip water studies by (Baldini et al., 2012), but drip waters in the present study demonstrate that typical "colloidal" elements show high frequency "events" (spiky signals in Fig. 4) and are independent of each other (Table A3). Consequently, our data from this system are not consistent with synchronous enrichment in these elements in drip water as a cause for synchronous enrichment in speleothem (e.g. Borsato et al., 2007). Nevertheless, more events of colloidal element

spikes during the wet season are observed which could also be related to the decomposition of biomass producing more organic colloids (Baldini et al., 2012). However, drip water might not be the main contributor of those elements, and they may be more controlled by deposition of particulates by air transport (e.g. dust during ventilation events, in-cave activity or block fall; Fairchild et al., 2010). Particulates could be trapped by the wet stalagmite surface and become incorporated into the stalagmite with calcite precipitation. Alternatively, in stalagmites with slow growth, a concentrating effect could enrich these elements.

Although we do not study farmed calcite samples here, we propose that the calcite growth rate (or growth cessations) in relation to the drip water (or dust) derived deposition can influence element incorporation in the stalagmite. During growth cessations those TE accumulate (sticking) on the surface of a stalagmite and are incorporated into calcite when stalagmite growth restarts. A synchronous enriched layer emerges when analyzing TE in stalagmites (e.g. by LA-ICP-MS methods; Sliwinski and Stoll, 2021; Borsato et al., 2007). Future investigations combining colloidal TE in drip water and farmed calcite would improve the

understanding of the calcite incorporation of such elements.



## 6. Conclusion

Over 16 months, monitoring of La Vallina cave sampled cave air ($pCO_2$ and $\delta^{13}C_{air}$) and cave conditions (temperature, drip rates) as well as drip water chemistry ($\delta^{13}C_{DIC}$, $\delta^{18}O$ and $\delta D$, 16 trace elements) and bedrock chemistry. The main findings include:

1. Although drip rates in the summer season of low rainfall and high evapotranspiration dropped to about 20% of the maximum winter season drip rate, the limited temporal variation in drip water $\delta^{18}O$ and $\delta D$ suggests a moderately- to well-mixed karst water reservoir in nearly all sites averaged by rainfall composition during recharge phases (positive P-PET). Isotopic offsets among sites are attributed to varying significance of evaporation in the soil and karst prior to entering the discrete water reservoir and different transit times of water through the discrete reservoirs.

2. The carbon isotopic ratio of drip water ($\delta^{13}C_{DIC}$) is lowest in summer and autumn and is likely controlled by biological processes of the covering vegetation and soil (root respiration, microbial activity). Temperature may be the dominant driver of vegetation, soil $pCO_2$ and $\delta^{13}C_{DIC}$ on the seasonal scale, but additional factors such as soil moisture and degradation of leaf matter in autumn enhancing microbial activity may also contribute to $\delta^{13}C_{DIC}$ variation. We do not find evidence for significant changes in the degree of open vs. closed system dissolution on the basis of $^{14}C$ drip
water measurements. Potential equilibration with cave air cannot be excluded.

3. Elements congruently dissolved from the host bedrock suggest higher degrees of bedrock dissolution in summer and autumn months of higher soil $pCO_2$. We find limited significance of PCP in most drip water locations, because the season of lowest cave $pCO_2$ is also the season of highest drip rate, so two key drivers of PCP feature opposing seasonal trends. Estimation of speleothem growth rate using monitoring data suggests that some drip sites may precipitate
$CaCO_3$ in only one part of the year, whereas other locations may feature more continuous growth.

4. Other sources of elements substantially enriched compared to bedrock composition are discussed. We do not find a synchronous autumnal peak in concentration of insoluble (colloidal and detrital) trace elements nor a correlation between their concentration and drip rate, although maximum concentrations in many of these elements generally occur in the wetter season. Synchronous peaks in such elements in stalagmites may therefore reflect other processes
related to seasonal minima in speleothem growth or additional deposition from directly advected aerosols.

The combination of monitoring cave conditions, hydrology and drip water chemistry clarifies processes significant to interpretation of speleothem records in this and other seasonally ventilating midlatitude cave systems.

## Data availability

Additional information is provided in the Appendix to this manuscript.





**Author contribution**

OK, JS and HS conceived and designed the research; HS supervised the study; OK, SG, LR and HS collected the samples; OK, LE and NH performed the lab analysis; OK, JS and HS wrote and edited the paper, all co-authors reviewed the manuscript.

**Competing interests**

The authors declare that they have no conflict of interest.

**Acknowledgements**

This work was supported by ETH Zürich [grant number ETH-1318-1]. We thank Madalina Jaggi for analytical support of $\delta^{13}C_{DIC}$ and $\delta^{18}O_{dripwater}$. We thank Pauline Treble and Jasper Wassenburg for valuable inputs improving the manuscript.




## Appendix

**Table A1: Limit of quantification (LOQ) of elements measured on an Agilent 8800 QQQ-ICP-MS at ETH Zürich. To detect the LOQ a series of dilutions of a standard solution was measured. Elements of samples not yielding sufficient signal above LOQ are listed in italics. Fig. 6 in the manuscript shows data only above LOQ.**

| Limit of quantification (LOQ) | Mass/mode | µg g$^{-1}$ |
|---|---|---|
| Li | 7 / No Gas | $<1*10^{-6}$ |
| Na | 23 / He | $<1*10^{-3}$ |
| Mg | 25 / H$_2$ | $<1*10^{-3}$ |
| Al | 27 / No gas | $1*10^{-4}$ |
| Si | 28 (44*) / O$_2$ | $1*10^{-2}$ |
| *P* | *31 (47*) / O$_2$* | *$1*10^{-2}$* |
| S | 32 (48*) / O$_2$ | $2.5*10^{-2}$ |
| K | 39 / H$_2$ | $<1*10^{-3}$ |
| Ca | 43 / No gas | $<1*10^{-2}$ |
| Cr | 53 / No gas | $1*10^{-5}$ |
| Mn | 55 / H$_2$ | $<1*10^{-5}$ |
| *Fe* | *56 / H$_2$* | *$1*10^{-3}$* |
| *Co* | *59 / H$_2$* | *$<1*10^{-5}$* |
| *Ni* | *62 / No gas* | *$1*10^{-4}$* |
| Cu | 65 / No gas | $<1*10^{-5}$ |
| *Zn* | *66 / No gas* | *Not detected* |
| As | 75 / No gas | $2.5*10^{-5}$ |
| Sr | 86 / No gas | $<1*10^{-4}$ |
| Y | 89 / No gas | $<1*10^{-6}$ |
| *Cd* | *111 / No gas* | *$1*10^{-5}$* |
| Ba | 138 / No gas | $2.5*10^{-4}$ |
| U | 238 / No gas | $<1*10^{-6}$ |

*mass shift (+16) of second quadrupole after reaction with $^{16}$O in reaction cell


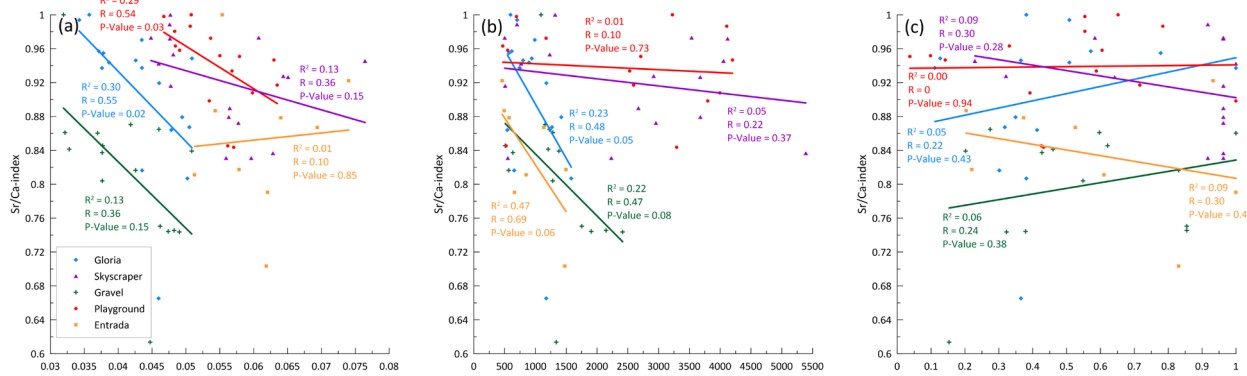

**Fig. A1: Checking potential driver of PCP indicated by Sr/Ca-index. Weak relationships between Sr/Ca-index and a) the Sr concentration (representing soil CO₂) and b) cave air pCO₂ (Gloria and Gravel) emerge. However, c) drip rate is hardly related to the Sr/Ca index. A stronger CO₂ gradient between drip water and air in contact with drip water increases PCP potential. Also, longer exposure time at slower drip rates might increase degassing, thus favor PCP.**





**Table A2: TE/Ca ratios of bedrock samples from La Vallina Cave. Note that Mg/Ca and Mn/Ca are given in mmol/mol whereas all other elements are given in µmol/mol.**

| | Li/Ca [µmol/mol] | Na/Ca [µmol/mol] | Mg/Ca [mmol/mol] | Al/Ca [µmol/mol] | Si/Ca [µmol/mol] | S/Ca [µmol/mol] | K/Ca [µmol/mol] | Cr/Ca [µmol/mol] | Mn/Ca [mmol/mol] | Cu/Ca [µmol/mol] | As/Ca [µmol/mol] | Y/Ca [µmol/mol] | Ba/Ca [µmol/mol] | U/Ca [µmol/mol] | Sr/Ca [µmol/mol] |
|---|---|---|---|---|---|---|---|---|---|---|---|---|---|---|---|
| La Vallina 1 | 2.92 | 53.49 | 3.25 | 135.21 | 113.31 | 196.70 | 57.44 | 0.26 | 0.54 | 0.95 | 0.17 | 4.53 | 1.41 | 0.86 | 431.95 |
| La Vallina 2 | 7.66 | 124.44 | 6.49 | 164.23 | 103.68 | 547.67 | 174.24 | 0.18 | 0.29 | 2.34 | 0.44 | 3.56 | 1.67 | 0.97 | 328.20 |
| La Vallina 3 | 1.81 | 48.62 | 4.10 | 301.28 | 257.12 | 183.67 | 266.25 | 0.18 | 0.39 | 2.15 | 0.50 | 3.67 | 2.14 | 0.54 | 375.51 |
| La Vallina 4 | 4.94 | 142.41 | 5.92 | 264.49 | 209.49 | 424.63 | 169.22 | 0.47 | 0.29 | 2.26 | 0.32 | 4.10 | 1.74 | 0.65 | 368.09 |
| La Vallina 5 | 4.93 | 129.37 | 5.30 | 272.33 | 108.10 | 293.10 | 98.96 | 1.05 | 0.33 | 1.33 | 0.37 | 4.07 | 1.59 | 0.92 | 419.64 |
| La Vallina 6 | 4.37 | 103.84 | 6.03 | 143.72 | 62.03 | 248.80 | 119.88 | 0.24 | 0.25 | 1.53 | 0.42 | 8.22 | 0.77 | 0.79 | 417.75 |
| La Vallina 7 | 8.51 | 197.90 | 6.92 | 741.52 | 904.02 | 371.72 | 91.40 | 0.41 | 0.17 | 1.49 | 0.10 | 3.50 | 4.24 | 1.60 | 1090.95 |
| La Vallina 8 | 7.58 | 144.47 | 6.27 | 277.57 | 72.99 | 330.28 | 65.87 | 0.34 | 0.31 | 1.59 | 0.33 | 6.78 | 1.47 | 0.81 | 507.75 |
| La Vallina 9 | 3.57 | 69.76 | 6.10 | 131.62 | 73.60 | 217.24 | 91.03 | 0.30 | 0.34 | 2.30 | 0.38 | 8.37 | 1.41 | 1.06 | 459.47 |
| La Vallina 10 | 3.97 | 144.57 | 5.19 | 628.81 | 1142.11 | 245.40 | 101.72 | 0.25 | 0.34 | 2.23 | 0.31 | 10.00 | 1.53 | 0.65 | 424.82 |
| La Vallina 11 | 4.57 | 556.62 | 5.20 | 337.39 | 357.75 | 398.32 | 2822.13 | 0.94 | 0.37 | 10.43 | 0.33 | 5.29 | 1.42 | 0.70 | 294.05 |
| La Vallina 12 | 6.66 | 130.59 | 5.30 | 176.52 | 46.67 | 353.34 | 220.38 | 0.20 | 0.40 | 1.02 | 0.08 | 3.64 | 3.16 | 2.23 | 650.99 |
| La Vallina 13 | 6.90 | 134.62 | 5.72 | 332.14 | 183.58 | 307.65 | 80.88 | 0.75 | 0.30 | 1.03 | 0.20 | 9.63 | 1.28 | 0.96 | 498.33 |
| La Vallina 14 | 1.66 | 91.72 | 37.38 | 148.38 | 150.18 | 179.31 | 346.91 | 0.87 | 1.12 | 3.28 | 1.18 | 7.70 | 1.43 | 0.70 | 88.30 |
| La Vallina 15 | 4.34 | 115.88 | 5.11 | 322.70 | 314.12 | 393.50 | 428.13 | 0.97 | 1.25 | 4.28 | 0.24 | 16.71 | 2.00 | 1.67 | 309.57 |
| La Vallina 16 | 5.20 | 60.53 | 5.24 | 387.53 | 485.96 | 539.30 | 275.88 | 1.96 | 1.83 | 3.71 | 0.99 | 19.54 | 5.86 | 1.92 | 362.93 |
| La Vallina 17 | 2.31 | 71.37 | 4.01 | 162.99 | 129.30 | 291.31 | 277.49 | 1.14 | 1.39 | 3.62 | 0.40 | 9.47 | 1.61 | 1.21 | 360.33 |
| La Vallina 18 | 3.47 | 44.51 | 5.44 | 186.74 | 131.41 | 426.59 | 109.53 | 1.30 | 1.39 | 1.24 | 0.38 | 10.89 | 1.36 | 1.19 | 462.13 |
| La Vallina 19 | 6.22 | 157.43 | 6.33 | 152.84 | 97.07 | 580.71 | 595.02 | 1.16 | 1.87 | 4.67 | 0.19 | 12.51 | 2.63 | 1.04 | 352.99 |
| La Vallina 20 | 0.06 | 25.56 | 2.25 | 47.76 | 10.17 | 11.21 | 99.95 | 0.07 | 0.00 | 1.18 | 0.04 | 0.75 | 0.41 | 0.09 | 35.91 |
| La Vallina 21 | 5.36 | 144.38 | 6.56 | 138.71 | 87.94 | 556.32 | 582.05 | 0.88 | 1.34 | 3.97 | 0.30 | 5.27 | 2.20 | 2.22 | 423.48 |
| La Vallina 22 | 3.75 | 287.48 | 2.68 | 157.90 | 146.49 | 230.80 | 1380.08 | 1.80 | 2.40 | 15.80 | 0.15 | 5.44 | 6.52 | 1.47 | 486.78 |
| La Vallina 23 | 1.83 | 95.17 | 2.26 | 238.06 | 234.60 | 441.57 | 473.19 | 0.94 | 2.87 | 5.37 | 0.24 | 4.09 | 1.66 | 1.06 | 392.96 |
| La Vallina 24 | 4.30 | 89.27 | 4.54 | 116.73 | 409.53 | 420.07 | 337.29 | 0.45 | 1.06 | 4.04 | 0.24 | 4.50 | 1.38 | 1.22 | 368.05 |
| La Vallina 25 | 2.50 | 72.15 | 2.50 | 125.90 | 89.50 | 250.21 | 417.33 | 1.01 | 2.22 | 1.79 | 0.13 | 3.18 | 0.90 | 1.10 | 346.26 |
| La Vallina 26 | 0.74 | 40.08 | 6.76 | 91.87 | 78.63 | 205.23 | 219.66 | 0.36 | 1.02 | 21.15 | 0.39 | 3.68 | 0.82 | 0.51 | 167.60 |
| La Vallina 27 | 2.03 | 42.13 | 2.91 | 254.22 | 342.34 | 251.77 | 231.77 | 1.86 | 1.76 | 2.30 | 0.45 | 4.76 | 1.62 | 1.37 | 253.82 |
| La Vallina 28 | 2.61 | 64.21 | 3.35 | 199.76 | 199.41 | 258.28 | 271.88 | 0.90 | 1.70 | 2.89 | 0.42 | 4.80 | 3.07 | 1.77 | 631.91 |
| La Vallina 29 | 1.83 | 108.66 | 2.43 | 286.90 | 349.23 | 250.79 | 516.00 | 1.24 | 1.89 | 8.61 | 0.34 | 6.05 | 1.97 | 1.82 | 383.90 |
| La Vallina 30 | 1.05 | 33.05 | 1.85 | 204.01 | 215.94 | 134.20 | 238.86 | 0.62 | 2.86 | 2.75 | 0.33 | 5.46 | 3.55 | 1.18 | 483.73 |
| La Vallina 31 | 5.42 | 78.64 | 9.02 | 146.10 | 33.11 | 230.19 | 26.67 | 0.40 | 0.33 | 0.66 | 0.17 | 3.94 | 1.47 | 1.09 | 656.69 |
| La Vallina 32 | 6.53 | 106.20 | 10.00 | 184.81 | 53.89 | 279.69 | 98.23 | 0.61 | 0.29 | 0.98 | 0.22 | 4.02 | 1.71 | 1.14 | 660.10 |
| La Vallina 33 | 3.02 | 156.97 | 2.58 | 430.44 | 1192.56 | 265.52 | 344.14 | 1.42 | 0.40 | 1.56 | 0.11 | 4.66 | 1.78 | 0.70 | 509.78 |
| La Vallina 34 | 2.99 | 177.67 | 2.32 | 1580.78 | 2637.75 | 292.56 | 1057.53 | 4.62 | 1.47 | 4.71 | 2.64 | 6.49 | 2.66 | 0.78 | 531.51 |
| La Vallina 35 | 9.69 | 570.39 | 4.22 | 540.79 | 763.64 | 1233.64 | 2087.39 | 0.51 | 0.27 | 6.86 | 0.30 | 8.93 | 1.78 | 0.54 | 378.38 |
| La Vallina 36 | 5.93 | 369.99 | 3.03 | 749.58 | 1111.95 | 809.85 | 909.20 | 0.84 | 0.44 | 3.29 | 0.24 | 6.42 | 1.31 | 0.79 | 552.70 |
| La Vallina 37 | 5.77 | 157.85 | 2.97 | 1243.14 | 1924.64 | 478.18 | 559.88 | 5.33 | 0.77 | 2.84 | 3.06 | 7.00 | 4.26 | 0.77 | 582.26 |
| La Vallina 38 | 3.57 | 105.65 | 4.09 | 713.44 | 1284.40 | 333.68 | 162.19 | 1.70 | 1.98 | 3.43 | 0.53 | 4.53 | 2.18 | 0.91 | 477.62 |
| La Vallina 39 | - | 26.21 | 4.68 | 61.73 | 53.26 | 114.56 | 115.52 | 0.09 | 1.07 | 1.15 | 0.07 | 2.76 | 1.37 | 0.15 | 120.41 |
| La Vallina 40 | 2.10 | 123.09 | 2.97 | 382.28 | 653.75 | 257.92 | 217.61 | 0.36 | 1.63 | 2.73 | 0.43 | 7.81 | 1.19 | 1.12 | 609.93 |
| La Vallina 41 | 2.23 | 40.08 | 2.40 | 103.55 | 85.13 | 354.44 | 58.84 | 0.41 | 1.96 | 1.14 | 0.21 | 4.79 | 1.15 | 0.85 | 503.04 |
| La Vallina 42 | 1.32 | 96.81 | 2.50 | 113.94 | 128.07 | 222.60 | 564.04 | 0.24 | 2.37 | 1.72 | 0.14 | 3.42 | 2.23 | 0.69 | 602.16 |
| La Vallina 43 | 4.12 | 45.95 | 4.77 | 99.63 | 52.28 | 660.60 | 31.46 | 0.50 | 1.49 | 1.87 | 0.18 | 4.69 | 0.89 | 1.83 | 337.71 |
| La Vallina 44 | 5.67 | 113.76 | 6.01 | 722.82 | 1345.36 | 399.21 | 59.29 | 0.39 | 1.56 | 1.43 | 0.12 | 5.15 | 3.54 | 0.78 | 491.30 |
| La Vallina 45 | 1.86 | 195.87 | 2.67 | 98.18 | 92.14 | 205.73 | 736.68 | 0.60 | 1.86 | 3.89 | 0.26 | 10.34 | 1.91 | 1.09 | 555.85 |
| Mean | 4.04 | 130.86 | 5.28 | 317.80 | 411.52 | 348.40 | 404.20 | 0.91 | 1.12 | 3.55 | 0.42 | 6.34 | 2.05 | 1.06 | 438.83 |
| 2 SD | 4.38 | 228.03 | 10.36 | 607.04 | 1101.51 | 400.11 | 1060.13 | 2.01 | 1.59 | 7.63 | 1.13 | 7.08 | 2.46 | 0.95 | 346.31 |
| RSD [%] | 54.76 | 88.11 | 99.17 | 96.58 | 135.35 | 58.07 | 132.62 | 111.26 | 71.95 | 108.80 | 134.93 | 56.50 | 60.68 | 45.72 | 39.90 |






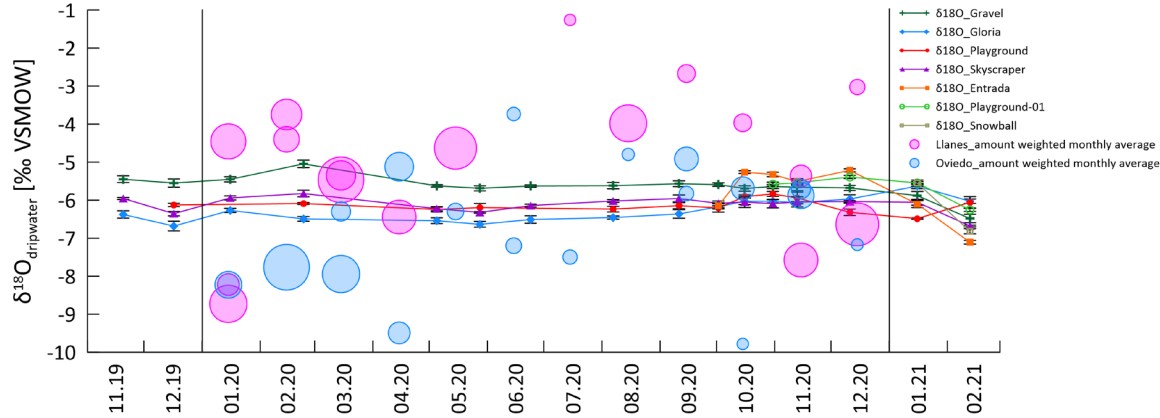

**Fig. A2: Drip water δ¹⁸O compared with two independent rainfall monitoring studies from NW Spain (Moreno et al., 2021; Stoll et al., 2015). The bubble size represents rain amount and the δ¹⁸O values are rainfall amount weighted monthly averages. Because these monitorings were performed in other years (2006-2009 and 2015-2016) the monthly averages are plotted on top of the year 2020 (marked with vertical lines). A seasonal trend is visible in rainfall with less negative δ¹⁸O values in summer months (and less rain) although variations are big between different years and months. The drip water δ¹⁸O signal supports the statement of a rather well-mixed drip water recharge reservoir. There can be multiple circles (pink and blue) because these rain monitorings were sampled for more than one seasonal cycle.**


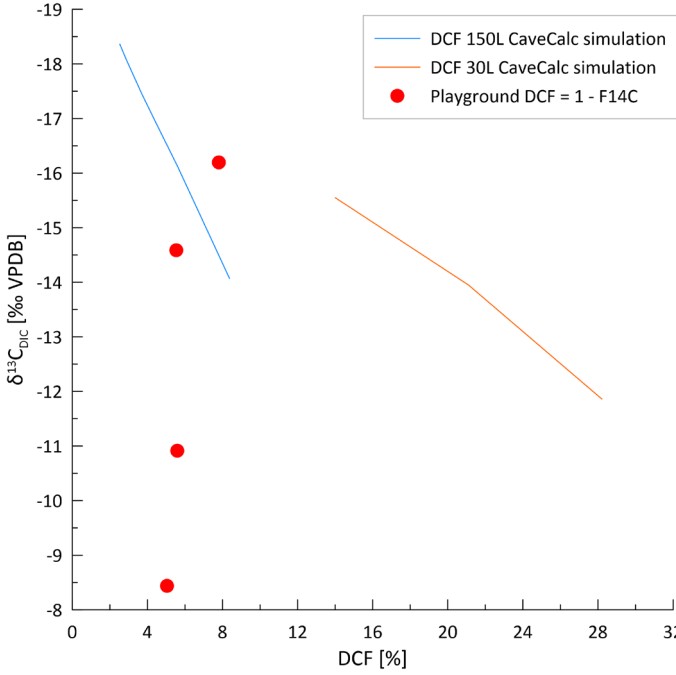


**Fig. A3: DCF calculated from fraction modern (F¹⁴C) (red dots) in contrast to CaveCalc simulations with different gas volumes representing different open (150L) vs. closed (30L) system conditions. Consequently, the measured range of DCF (5-8%) does not explain the range in δ¹³C_DIC.**






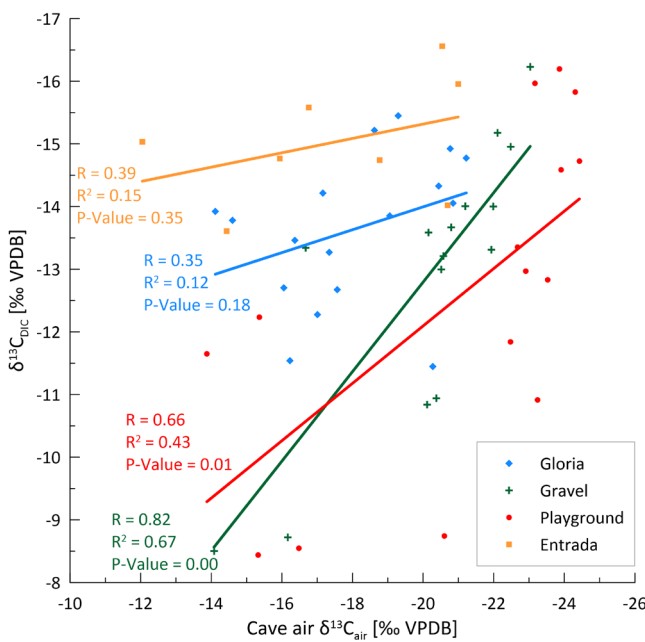

**Fig. A4: Correlation of drip water δ¹³C_DIC and cave air δ¹³C_air to test potential gas exchange with cave air. Pearson correlation coefficient (R), R² and p-value suggest reasonable correlations for Playground and Gravel but not for Entrada and Gloria.**

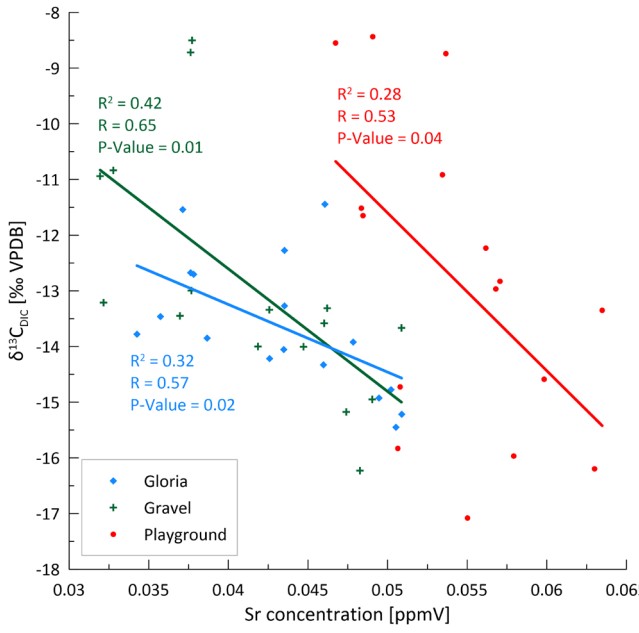

**Fig. A5: Sr concentration is inversely correlated with δ¹³C_DIC. This is interpreted as a bedrock dissolution signal since higher soil
pCO₂ with microbial (negative) imprint driven by temperature results in stronger dissolution.**





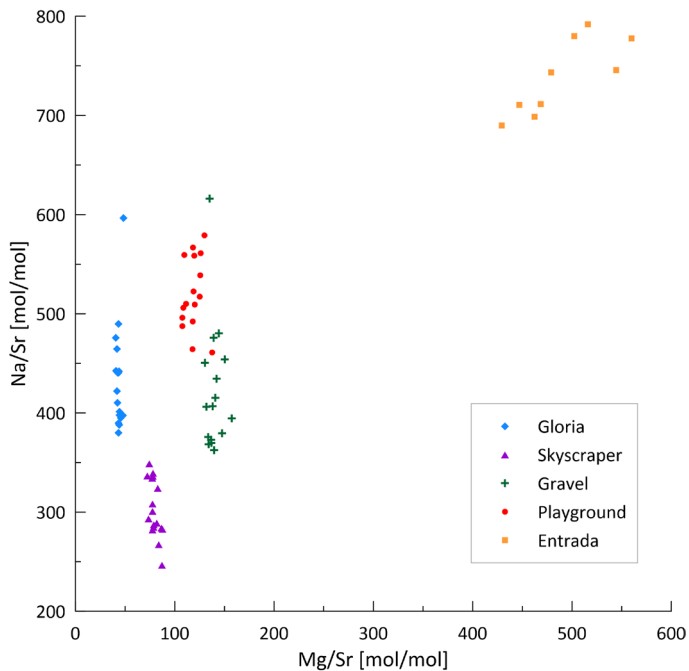

**Fig. A6: Mg/Sr vs. Na/Sr scatter plot highlighting the likely marine aerosol contribution at Entrada as a positive correlation emerges. Drip sites deeper in the cave do not indicate a relationship between Na and Mg, thus are not affected by marine aerosols.**

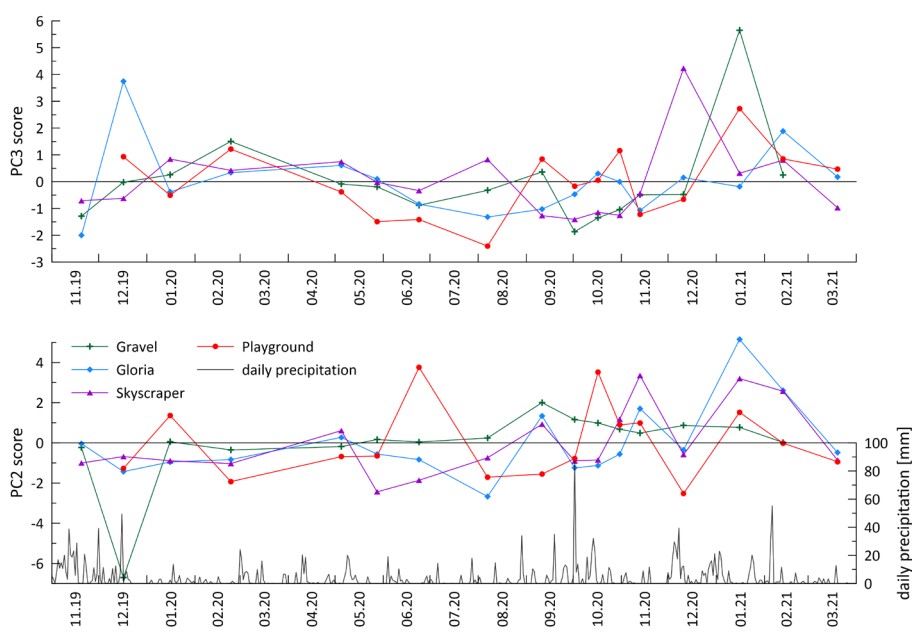


**Fig. A7: Temporal evolution of PC 2 and PC 3 score. PC 2 might be related to wash-in events (plotted with daily precipitation) since PC 2 elements are mostly spiky on event time scales (Fig. 6).**





**Table A3: Correlation table of typical "colloidal" elements with Y (another colloidal element) to show that no clear correlation** 800 **emerges. Pearson coefficients are statistically insignificant for numbers in italics (p-value > 0.05) whereas normal numbers are statistically significant (p-value < 0.05). However, in drip water correlations are weak and no trends are observed what contradicts the findings in stalagmites with synchronous change and high correlation.**

| Correlation with Y | Li | Al | K | Cr | Mn | Cu | As |
|---|---|---|---|---|---|---|---|
| Gravel | 0.08 | 0.01 | -0.03 | -0.19 | *0.27* | 0.07 | -0.78 |
| Gloria | 0.15 | 0.06 | 0.10 | 0.12 | 0.45 | 0.27 | 0.05 |
| Skyscraper | *-0.35* | 0.36 | -0.18 | 0.27 | *-0.07* | 0.02 | 0.36 |
| Playground | 0.25 | 0.55 | 0.03 | 0.18 | 0.82 | *0.20* | 0.16 |
| Entrada | *0.32* | 0.63 | 0.62 | -0.30 | 0.71 | 0.49 | 0.03 |

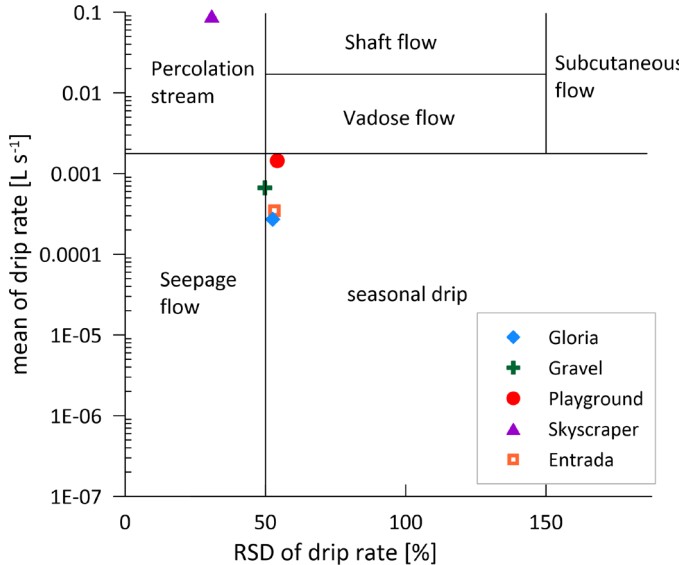


**Fig. A8: Drip classification after Baker et al. (1997). Most drips are seasonal close to the border to seepage flow. Only Skyscraper (behaving like a shower) is behaves like a percolation stream. Since point sampling only allowed to determine the instantaneous drip rate on the day of sampling a higher RSD is possible (minimum and maximum might not be recorded).**



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
