# Peer review of "Relationship of seasonal variations in drip water $\delta^{13}C_{DIC}$ , $\delta^{18}O$ and trace elements with surface and physical cave conditions of La Vallina Cave, NW Spain"

_Hydrology and Earth System Sciences, 2022_

## Referee Comment (RC1)

**Review of Kost et al. "Relationship of seasonal variations in drip water $d^{13}C_{DIC}$, $d^{18}O$ and trace elements with surface and physical cave conditions of La Vallina Cave, NW Spain" submitted to Hydrology and Earth System Sciences – hess-2022-386**

The manuscript by Kost et al. presents cave monitoring data of air and drip water including element concentrations and isotope data of La Vallina Cave, NW Spain. The data are very well presented and their interpretation is concise. I just have a few suggestions for improving the manuscript and thus, recommend minor revisions.

**General comments:**

In my opinion, the statement that speleothem precipitation is restricted to summer AND winter seems a bit contradicting. For example, $\delta^{13}C_{DIC}$ values and $pCO_2$ cave air concentrations (thus, cave ventilation pattern) hint at preferred precipitation in winter. How valid are the model data for this? Or do you mean at some locations in the cave, there is summer precipitation and in other cave locations, there is winter precipitation? Especially in the abstract, this is quite confusing. Probably needs a more detailed explanation in the abstract or a rewriting, cause in the conclusions, it is far better explained.

Methods: Please provide the information on which instrumentation was used for measuring Cave air temperature and relative humidity as well as the precision of the device(s). What is the measurement uncertainty of the Picarro for $pCO_2$ and $\delta^{13}C_{CO2}$?

Results section: There are already some interpretations /discussions occurring here: Cave air $CO_2$, hydrological conditions, 4.4 Isotopic composition of the drip water. Please go through that section and thoroughly separate what is a result and what is interpretation/discussion and move those parts to the discussion section.

For results section on PCP: I suggest to use also Mg/Ca and Ba/Ca ratios to help determine variations in PCP. The PCA shows that Mg, Ca, Sr and Ba seem to be influenced by the same environmental factor (PC1). Thus, I suggest including those ratios in the PCP part at least for Gloria, Skyscraper, Playground and Gravel drip water sites, which seem less influenced by seawater aerosol input. Also, what about a Sinclair plot?

Lines 545-570: I miss correlations between $\delta^{13}C_{DIC}$ and cave air $CO_2$ concentrations in this section. In figure 2, $\delta^{13}C_{DIC}$ and $pCO_2$ concentrations seem to show anti-correlations for the different sites. Please check, if that is the case. Thus, this will highly strengthen your argumentation that the degree of degassing (which depends on $CO_2$ concentrations of cave air) has an influence on $\delta^{13}C_{DIC}$ values. Revise text here and elsewhere.

I think you use "epikarst" when you mean the karst zone. Epikarst is just the uppermost zone of the karst, which is in contact with the soil. You cannot use it to describe the whole bedrock above the cave. Please clarify this. See for example Fairchild and Baker (2012) and Bakalowicz (2012).

**Further comments:**
Lines 19-20: This is a bit misleading. Please write instead: The carbon isotope signature of dissolved inorganic carbon of drip water…
Line 23: What kind of cave air measurements? Be more precise.

Line 124: What does the abbreviation DEM stand for?

Lines 154 and 203: ConFlo IV

Line 296, 316 and elsewhere: Please write hydrogen isotopic composition. When speaking of deuterium, only the hydrogen isotope with the mass 2 is meant.

Line 411: Also due to the good ventilation in winter diluting and removing the $CO_2$ degassed from drip water in winter. I suggest adding that factor a bit more in the discussion in this part.

---

## Author Comment (AC1)

RC 1:
The manuscript by Kost et al. presents cave monitoring data of air and drip water including element concentrations and isotope data of La Vallina Cave, NW Spain. The data are very well presented and their interpretation is concise. I just have a few suggestions for improving the manuscript and thus, recommend minor revisions.

**General comments:**
In my opinion, the statement that speleothem precipitation is restricted to summer AND winter seems a bit contradicting. For example, $\delta^{13}C_{DIC}$ values and $pCO_2$ cave air concentrations (thus, cave ventilation pattern) hint at preferred precipitation in winter. How valid are the model data for this? Or do you mean at some locations in the cave, there is summer precipitation and in other cave locations, there is winter precipitation? Especially in the abstract, this is quite confusing. Probably needs a more detailed explanation in the abstract or a rewriting, cause in the conclusions, it is far better explained.

[We thank the reviewer for this comment. To be clearer in what our findings show, we revised this passage in the abstract. The used model (I-STAL; Stoll et al., 2012) is well established and fed with monitoring data (Ca concentration, cave air $pCO_2$, temperature, drip rate). The findings of growth cessation in spring and fall leads to two growth phases during the year (winter AND summer) at most sites when conditions are favorable for calcite precipitation. So, growth conditions are favorable in summer and winter mainly driven by low cave $pCO_2$ in winter (ventilated conditions) and high oversaturation of dripwater in summer (high soil $pCO_2$ enhancing bedrock dissolution). The modelled calcite precipitation shows differences between sampling sites depending on the input parameters. One would expect rather similar growth behavior at Gloria, Gravel and Skyscraper (two growth phases in winter AND summer with growth cessation in spring and fall) and a single growth phase in winter at Playground. The example of Playground shows that irregular growth is possible in La Vallina cave since potential calcite precipitation at Playground is expected only during winter 20/21. Hence, the actual growth of a stalagmite growing in the cave depends on its location and cave/dripwater conditions. The growth cessation in fall is likely related to strong PCP effects as shown in Fig. 6 with increased Sr/Ca. In spring, low soil $CO_2$ and increasing cave $pCO_2$ restricts calcite precipitation. The negative growth rates during these phases could even suggest calcite dissolution.]

Methods: Please provide the information on which instrumentation was used for measuring Cave air temperature and relative humidity as well as the precision of the device(s). What is the measurement uncertainty of the Picarro for $pCO2$ and $\delta^{13}C_{CO2}$?

[We appreciate the request on cave air measurement instrument and its precision. A transparent reporting of instruments and uncertainties corresponds with our reporting philosophy. We now provide information on the instrument and its measurement accuracies used to measure humidity and temperature. Furthermore, we added measurement uncertainties for the Picarro used to measure the cave air $pCO2$ and $\delta^{13}C$. The figures already showed error bars but we missed to add the information in written format in the previous version.]

Results section: There are already some interpretations /discussions occurring here: Cave air $CO_2$, hydrological conditions, 4.4 Isotopic composition of the drip water. Please go through that section and thoroughly separate what is a result and what is interpretation/discussion and move those parts to the discussion section.

[We identified several parts of discussion elements in the results section and carefully tried to remove them from the results section and incorporate them into the discussion section. We have made our best effort to present results without significant interpretation, in a few gray areas the later results sections require brief explanation of the rationale (e.g. Sr/Ca-index as a PCP indicator or the principle of cave ventilation) and we retain these brief sentences for clarity.]

For results section on PCP: I suggest to use also Mg/Ca and Ba/Ca ratios to help determine variations in PCP. The PCA shows that Mg, Ca, Sr and Ba seem to be influenced by the same environmental factor (PC1). Thus, I suggest including those ratios in the PCP part at least for Gloria, Skyscraper, Playground and Gravel drip water sites, which seem less influenced by seawater aerosol input. Also, what about a Sinclair plot?
[As suggested by the reviewer, a Sinclair plot is now added to the supplementary figures, and cited in the main text section 4.5. This is the best argument for a similar control by PCP. For some sites like Playground the range in Mg/Ca is too low and no statistically significant slope was found (p-value too large). Therefore, we only report slopes (and slope error = SE) of statistically significant data sets. We realize generally slightly smaller slopes (mostly within error though) than suggested by Sinclair et al. (2012) or Wassenburg et al. (2020); however, this can be explained with varying partitioning coefficients. A paper in prep. discussing effects of partitioning coefficients on the slopes is in the pipeline.
As Fig. 4 (or the figure below) shows, Ba concentrations indicate a higher variability suggesting other factors additionally controlling Ba (e.g. detrital particles). However, the long-term trends, as suggested by PC1, are fairly similar to Sr/Ca- and Mg/Ca-index.
To keep the manuscript slim and avoid further discussion (there's a lot of data already and reviewer #2 prefers a shorter version), we decided to stick to Sr/Ca in the main text only but now mention that Mg/Ca shows the same pattern. We do not comment on Ba as a PCP proxy.]

Lines 545-570: I miss correlations between $\delta^{13}C_{DIC}$ and cave air $CO_2$ concentrations in this section. In figure 2, $\delta^{13}C_{DIC}$ and $pCO_2$ concentrations seem to show anti-correlations for the different sites. Please check, if that is the case. Thus, this will highly strengthen your argumentation that the degree of degassing (which depends on $CO_2$ concentrations of cave air) has an influence on $\delta^{13}C_{DIC}$ values. Revise text here and elsewhere.
[We thank the reviewer for this input. Previously, we had not included the (anti-)correlation figure for the sake of keeping the large paper more concise. We add the figure to the supplement (Fig. A5b) and now mention in section 5.3:
"Additionally, the modest inverse correlation between drip water $\delta^{13}C_{DIC}$ and cave air $pCO_2$ in these sites is consistent with greater extent of degassing (and potentially PCP) during periods of low cave air $pCO_2$."]

I think you use "epikarst" when you mean the karst zone. Epikarst is just the uppermost zone of the karst, which is in contact with the soil. You cannot use it to describe the whole bedrock above the cave. Please clarify this. See for example Fairchild and Baker (2012) and Bakalowicz (2012).
[We thank the reviewer for this comment. We now use the more general word "karst" instead of "epikarst" to avoid any misunderstanding. It is mostly used in relation to soil/karst air and since in this case the $CO_2$ production by roots extends all the way to the cave (roots penetrate cave in some areas), we broaden our description to include the entire rock above the cave.]

**Further comments:**

Lines 19-20: This is a bit misleading. Please write instead: The carbon isotope signature of dissolved inorganic carbon of drip water…

[We agree with the reviewer that this phrase might be misleading for the reader. This is changed in the revised version.]

Line 23: What kind of cave air measurements? Be more precise.

[We added more specific information in brackets.]

Line 124: What does the abbreviation DEM stand for?

[A DEM is a digital elevation model. The full description is added now in the figure caption (Fig. 1).]

Lines 154 and 203: ConFlo IV

[Good call! A typo which is revised in the new version.]

Line 296, 316 and elsewhere: Please write hydrogen isotopic composition. When speaking of deuterium, only the hydrogen isotope with the mass 2 is meant.

[We totally agree with the reviewer's suggestion. We changed it throughout the manuscript accordingly.]

Line 411: Also due to the good ventilation in winter diluting and removing the $CO_2$ degassed from drip water in winter. I suggest adding that factor a bit more in the discussion in this part.

[Good point! We added this mechanism in the discussion.]

References

Sinclair, D. J., Banner, J. L., Taylor, F. W., Partin, J., Jenson, J., Mylroie, J., Goddard, E., Quinn, T., Jocson, J., and Miklavič, B.: Magnesium and strontium systematics in tropical speleothems from the Western Pacific, Chemical Geology, 294-295, 1-17, https://doi.org/10.1016/j.chemgeo.2011.10.008, 2012.

Wassenburg, J. A., Riechelmann, S., Schröder-Ritzrau, A., Riechelmann, D. F. C., Richter, D. K., Immenhauser, A., Terente, M., Constantin, S., Hachenberg, A., Hansen, M., and Scholz, D.: Calcite Mg and Sr partition coefficients in cave environments: Implications for interpreting prior calcite precipitation in speleothems, Geochimica et Cosmochimica Acta, 269, 581-596, https://doi.org/10.1016/j.gca.2019.11.011, 2020.

---

## Author Comment (AC2)

RC 2:

The manuscript submitted by Kost et al deals with the characterization of the atmosphere-karst vadose zone-cave system at a cave located in the Northwest Iberian Peninsula. The characterisation is based on a cave monitoring programme including measurements of cave air, bedrock chemistry, and drip water $\delta^{13}$C DIC, $\delta^{18}$O and $\delta$D as well as 16 trace elements with irregular measurements for about 16 months. The study finds sensitivities of drip rates to climatic seasonality but relatively low variations of the stable water isotopes which they attribute to a well-mixed reservoir in the vadose zone above the cave. $\delta^{13}$C DIC indicates the influence of seasonal vegetation dynamics and related microbial activity. Their measurements further indicate limited the extent of the seasonal variation of degassing and prior calcite precipitation (PCP). They also indicate that stalagmite growth is limited to the summer and winter seasons.

The study is well-written and concise. The simultaneous monitoring of multiple variable permits in-depth analysis of the processes controlling cave drip water composition and stalagmite growth in the observed cave. There are only few minor points that should be addressed:

- The results section already includes a lot of interpretation and link to other references, which should be moved to the discussion section.

[We identified several parts of discussion elements in the results section and carefully tried to remove them from the results section and incorporate them into the discussion section. We have made our best effort to present results without significant interpretation, in a few gray areas the later results sections require brief explanation of the rationale (e.g. Sr/Ca-index as a PCP indicator or the principle of cave ventilation) and we retain these brief sentences for clarity.]

- Assumption of P-PET equals available water for downward percolation from the surface. I do not completely agree with this assumption as it assumes that AET equals PET, which is often not the case and which may explain why the drips are also active in summer. AET and PET depend on factors like land cover type, soil thicknesses, and rooting depth. Consequently, they affect what is left for feeding the drips below and also how fast a drip reacts to rainfall events (see for instance Berthelin and Hartmann, 2020; Carrière et al., 2020; Sarrazin et al., 2018).

[We thank the reviewer for highlighting this point. We now clarify in section 4.3:
"We note that the calculated PET may overestimate the actual evapotranspiration (AET) and therefore underestimate actual water balance. Our purpose is to illustrate the trend in seasonal variation in water balance rather than a quantitative calculation of water balance; for this purpose PET is suitable and avoids the uncertainties of assigning a single crop coefficient over a landscape of heterogeneous vegetation cover. Recharge of the water reservoir potentially only occurs when water balance is positive (Fig. 3e)."
Additionally in section 5.2:
"Recently published hydrological models from semi-arid caves suggest a recharge threshold has to be overcome (saturated soil and epikarst) to contribute to the deeper water reservoir feeding the cave drip water (Markowska et al., 2016; Baker et al., 2021). Hence, when (P-PET) is negative in summer, any rain is likely evaporated and does not recharge the water reservoir. Even with a lower actual evapotranspiration (AET) summer recharge is reduced. However, the active drips in summer suggest a large water reservoir, in line with the well-mixed reservoir discussed below, which does not dry out during summer and keeps drips active even if recharge is minimal."

Active drip sites in summer might on one hand be fed by higher recharge as suggested by using AET instead of PET, on the other hand a water reservoir doesn't need to dry out if it's large enough even if effective recharge is zero. Our findings of well-mixed water reservoirs indicates relatively large reservoirs (no response to single events), therefore supports the argumentation that the water reservoir does not empty in summer keeping drip sites active. We hope to satisfy the reviewer by extending this discussion including AET (instead of PET) as a potential driver of active drip sites in summer.]

- The authors mention the 7-day antecedent cumulative rainfall method by Baker et al (2020, 2021) did not work but did they also try the simple bucket model used in the same studies? This may help resolving the question about PET being a goof proxy for AET and allow to estimate vertical percolation in the preceding years.

[We thank the reviewer for reminding us of the simple bucket model. The main limitation in applying hydraulic models is that we have data on driprate only at discrete points in time during the monthly sampling, not continuously. For example, the temporal resolution (monthly) does not allow to investigate response time to rainfall events. Given this limitation, and that hydraulic model was not a main focus of the study, since we are asked to shorten the manuscript, we decide not to explore further modeling approaches in this paper.]

- The authors state that the spatial variability of drip waters suggest that they are not feed by the same reservoir but multiple reservoirs with varying dominance of diffuse and preferential flow routes. However, different contributions of diffuse and preferential flow would not affect the time, volume weighted averages of the drips. More preferential flow would only result in stronger visibility of the seasonal isotopic signal of the rain, which the authors did not find. The obvious differences in average $\delta^{18}O$ and $\delta D$ must originate from different processes than mixing alone. Maybe, the processes found by Treble et al. (2022) can support the interpretation.

  When discussing the effect of evaporation on stable water isotopes, it is not clear whether the authors mean evaporation and/or transpiration because only the former results in fractionation.

[As we discuss in section 5.2, we do not have matched rainfall and $\delta^{18}O$ for the same monitoring period. Thus, we cannot rule out the possibility that inter site differences may reflect varying average transit times among drips, so that the amount weighted drip water $\delta^{18}O$ of different drips would converge over a suitably long time period which averaged interannual differences in amount weighted $\delta^{18}O$ of precipitation. If significant differences in mean transit times existed among drip sites, and the amount weighted average precipitation in the year prior to monitoring were more positive than that during the monitored season, then drips with slower average transit times might be offset to the previous year's value. Interannual variation of about 1.25‰ in the yearly precipitation amount weighted average $\delta^{18}O$ is seen among the available precipitation data (Fig. 9) which includes 15 years from GNIP station Santander in the period 2000-2015 (no recent data available; Iaea/Wmo, 2022) and individual rain events monitored near Oviedo from February 2015 to February 2016 (Moreno et al., 2021).

We also have expanded the citations for the statement that "Prior studies have shown that nearby drip sites can undergo very different routing in the unsaturated zone undergoing different evaporative fractionation (Treble et al., 2022; Markowska et al., 2016)."

To clarify the role of eucalyptus on evaporation, we further detail now in section 5.2: "The heterogenous coverage by deep rooted trees such as Eucalyptus may contribute to lateral variations in evaporative fractionation, because they can increase evaporation from the upper soil horizons via the hydraulic redistribution of deep groundwater to the upper soil (Brooksbank et al., 2011)."

"These effects may be more extreme now due to spatially variable vegetation and extreme evapotranspiration and hydraulic redistribution of perennially green Eucalyptus (Brooksbank et al 2011), compared to native deciduous oaks."

- Generally, the discussion section seems to be quite long, especially when considering that it already starts in the results section (see my previous comment). It would be helpful, if the authors could tailor it a bit more to the main outcomes of the paper.

[We appreciate the suggestion. From the two reviews, we found difficult to ascertain what portions should be adjusted without disrupting the clarity of the paper. For this reason, we have not shortened the paper, but have taken the caution to add the additional requested figures to the supplement rather than the main text.]

I am convinced that these points can be implemented within the frame of minor revisions.

References

Baker, A., Scheller, M., Oriani, F., Mariethoz, G., Hartmann, A., Wang, Z., and Cuthbert, M. O.: Quantifying temporal variability and spatial heterogeneity in rainfall recharge thresholds in a montane karst environment, Journal of Hydrology, 594, 125965, https://doi.org/10.1016/j.jhydrol.2021.125965, 2021.

IAEA/WMO: Global Network of Isotopes in Precipitation. The GNIP Database., 2022.

Markowska, M., Baker, A., Andersen, M. S., Jex, C. N., Cuthbert, M. O., Rau, G. C., Graham, P. W., Rutlidge, H., Mariethoz, G., Marjo, C. E., Treble, P. C., and Edwards, N.: Semi-arid zone caves: Evaporation and hydrological controls on δ18O drip water composition and implications for speleothem paleoclimate reconstructions, Quaternary Science Reviews, 131, 285-301, https://doi.org/10.1016/j.quascirev.2015.10.024, 2016.

Moreno, A., Iglesias, M., Azorin-Molina, C., Pérez-Mejías, C., Bartolomé, M., Sancho, C., Stoll, H., Cacho, I., Frigola, J., Osácar, C., Muñoz, A., Delgado-Huertas, A., Bladé, I., and Vimeux, F.: Measurement report: Spatial variability of northern Iberian rainfall stable isotope values – investigating atmospheric controls on daily and monthly timescales, Atmos. Chem. Phys., 21, 10159-10177, 10.5194/acp-21-10159-2021, 2021.

Treble, P. C., Baker, A., Abram, N. J., Hellstrom, J. C., Crawford, J., Gagan, M. K., Borsato, A., Griffiths, A. D., Bajo, P., Markowska, M., Priestley, S. C., Hankin, S., and Paterson, D.: Ubiquitous karst hydrological control on speleothem oxygen isotope variability in a global study, Communications Earth & Environment, 3, 29, 10.1038/s43247-022-00347-3, 2022.

---

## Author Response (AR2)

Dear Editor,

We thank you for the feedback on the revisions.

Regarding the instruction to shorten the manuscript, we have completed a sentence by sentence revision to make the text more concise, without sacrificing content or interpretation. This effort has shortened the manuscript by 584 words.

Regarding the suggestion to implement the bucket model of Baker et al (2020, 2021), we fully appreciate the value of quantitative modeling for geochemical and hydrological data and the value of the bucket model published by Baker et al (2020, 2021). Baker et al uses highly temporally resolved drip water estimates (15 minute resolution, during 6 years) to "determine recharge thresholds, and their spatiotemporal heterogeneity" (Baker et al., 2021).

In detail, the stated purpose of the bucket model is to estimate overflow capacity [mm] and a drainage parameter [mm/day]. The bucket model employs the precipitation and AET or PET as input. Then "The two free parameters are optimised on the observation of recharge within a 7-day window of a precipitation event, to minimise the difference between observed recharge events and modelled concentrated recharge." The use of the bucket model to estimate these parameters is contingent on time-resolved drip rate (recharge) measurements within this 7-day interval.

In our monitoring study, spanning 16 months, the median time interval between our instantaneous drip rate observations is 28 days, with three monitoring periods separated by 13 to 15 days. The rationale for choosing a 7-day window in the midlatitude site described in Baker et al., 2021 is also appropriate for our midlatitude setting (typical duration of precipitation events and their frequency due to frontal systems from the mid-latitude westerly zonal circulation). Because the focus of our study was on geochemical tracers rather than high resolution hydrological response, the data resolution collected in our observations is not sufficient to apply the bucket model as described in Baker et al (2020, 2021). While in some cases, there may be a rainfall event within 7 days of our sampling date, from a single measurement, we cannot distinguish if our sample corresponds to the peak or rising or falling limb of the discharge response. Our samplings do not occur at a standard time following the onset of a precipitation event, and it is very unlikely that our samplings would always correspond to the same relative position (e.g. rising limb) of the response.

We have revised the discussion on hydrographic response to explicitly mention the challenges in application of models to our resolution drip data:

"However, the long median interval between drip rate measurements in this study (28 days) complicates efforts to derive relationships between drip rate and rainfall or precipitation-evapotranspiration, and precludes the use of quantitative flow models such as bucket models to derive recharge thresholds and overflow capacity (Baker et al., 2020; Baker et al., 2021) since we cannot ascertain if our single estimation of drip rate reflects peak discharge or rising or falling limb of discharge after a precipitation event. For example, we did not identify a relationship between 7-day antecedent cumulative rainfall and drip rate, found in studies featuring higher resolution drip rate data (Baker et al., 2020; Baker et al., 2021). Continuous hydrographs recording drip rates or tracer experiments would be necessary to determine the response time to rain events and to define thresholds of rainfall amount affecting drip sites individually (Markowska et al., 2016)(Baker et al., 2020; Baker et al., 2021)."

Finally, we have substituted one new reference published since the manuscript was submitted, which includes growth rates of the active stalagmites in the same monitored cave, rather than two older references with ages of non-active stalagmites.

Kind regards,

Oliver Kost, Heather Stoll and co-authors